# Intracellular carbon storage by microorganisms is an overlooked pathway of biomass growth

Kyle Mason-Jones [1,2] ✉, Andreas Breidenbach[2,3], Jens Dyckmans[4], Callum C. Banfield [2,3] & Michaela A. Dippold [2,3] ✉

The concept of biomass growth is central to microbial carbon (C) cycling and ecosystem nutrient turnover. Microbial biomass is usually assumed to grow by cellular replication, despite microorganisms' capacity to increase biomass by synthesizing storage compounds. Resource investment in storage allows microbes to decouple their metabolic activity from immediate resource supply, supporting more diverse microbial responses to environmental changes. Here we show that microbial C storage in the form of triacylglycerides (TAGs) and polyhydroxybutyrate (PHB) contributes significantly to the formation of new biomass, i.e. growth, under contrasting conditions of C availability and complementary nutrient supply in soil. Together these compounds can comprise a C pool $0.19 \pm 0.03$ to $0.46 \pm 0.08$ times as large as extractable soil microbial biomass and reveal up to $279 \pm 72\%$ more biomass growth than observed by a DNA-based method alone. Even under C limitation, storage represented an additional 16–96% incorporation of added C into microbial biomass. These findings encourage greater recognition of storage synthesis as a key pathway of biomass growth and an underlying mechanism for resistance and resilience of microbial communities facing environmental change.

Microbial assimilation of organic resources is crucial to the flow of C and other nutrients through ecosystems. Soil heterotrophs perform key steps in terrestrial carbon (C) and nutrient cycles, yet how microorganisms use the available organic resources and regulate their allocation to competing metabolic demands remains a subject of research and debate[1–3]. Microbial assimilation of organic C into an organism is conceptualized as biomass growth. This is frequently understood as synonymous with an increase in individuals, in other words, the replicative growth of microbial populations. However, many microorganisms are capable of storage, defined as the accumulation of chemical resources in particular forms or compartments to secure their availability for future use by the storing organism[4]. Various microbial storage compounds are known, among them

polyhydroxybutyrate (PHB) and triacylglycerides (TAGs)[5,6]. These two C-rich storage compounds are of particular interest as they are accumulated by diverse microbial taxa[4] and methods are available for their measurement in soil[7,8]. These are both hydrophobic lipids that are stored as inclusion bodies in the cytosol (i.e., intracellular lipid droplets)[5]. PHB is a high-molecular-weight polyester of β-hydroxybutyrate, while TAGs consist of three fatty acids (of diverse structures) esterified to a glycerol backbone[4]. PHB storage is only known among bacteria, while TAGs are accumulated by both bacteria and fungi[4]. Biosynthesis of PHB has been demonstrated by compound-specific measurement in soil[8] and TAGs in marine and soil systems show responsiveness to resource supply consistent with a C-storage function[9,10]. Microbial storage could substantially influence microbial

[1]Department of Terrestrial Ecology, Netherlands Institute of Ecology (NIOO-KNAW), Wageningen, the Netherlands. [2]Biogeochemistry of Agroecosystems, Department of Crop Sciences, Georg-August University of Göttingen, Göttingen, Germany. [3]Geo-Biosphere Interactions, Department of Geosciences, University of Tübingen, Tübingen, Germany. [4]Centre for Stable Isotope Research and Analysis, Georg-August University of Göttingen, Göttingen, Germany. ✉e-mail: k.masonjones@nioo.knaw.nl; michaela.dippold@uni-tuebingen.de

fluxes of C and other nutrients[11], changing our understanding of soil biogeochemical fluxes and their response to environmental changes.

Biomass growth is a cornerstone concept at scales from the ecological stoichiometry of individual cells to microbially-explicit models of the C cycle[12,13], and for defining the nutrient demands of organisms and their productivity[12]. Accumulation of storage compounds corresponds to an increase in microbial biomass without replication, and therefore represents an alternative pathway for growth that is not usually considered in the C cycle. There is therefore a need to assess how severely the omission of storage may bias our understanding of C assimilation and utilisation[4]. Conventional measurement of soil microbial biomass uses fumigation with chloroform to lyse cells, followed by extraction of the released biomass into an aqueous solution for measurement (chloroform fumigation-extraction, CFE)[14]. This method assumes a proportionality between extractable and non-extractable biomass[15]. Other measures in widespread use are proxies such as cell membrane lipids or substrate-induced respiration[16–18]. Only CFE provides biomass in units of C, however, and these other methods are typically calibrated against it. However, hydrophobic storage compounds like PHB and TAGs are not extractable in aqueous solution and are therefore overlooked by CFE. Furthermore, there is no biological reason to expect proportionality between these storage compounds and any of the conventional biomass proxies. DNA-based measures of microbial abundance and replication also do not capture storage[19,20], since it is not expected to form a constant proportion of each cell's biomass.

Interpretation of microbial storage patterns is facilitated by distinguishing two storage modes, which represent the end-members on a gradient of storage strategies[4,21]. Surplus storage is the accumulation of resources that are available in excess of immediate needs, at little to no opportunity cost, while reserve storage accumulates limited resources at the cost of other metabolic functions. Surplus storage of C would be predicted under conditions of C oversupply, when replicative growth is constrained by other factors such as nutrient limitation. Reserve storage, on the other hand, indicates that storage may also occur under C-limited conditions. The evidence assembled from pure culture studies confirms the operation of both storage modes among microorganisms[4,22–24]. Here we experimentally investigate the importance of microbial storage in soil, and show how storage responses to resource supply and stoichiometry can advance our understanding of resource allocation and microbial biomass growth. We hypothesized as follows:

1. Microbial storage compounds are a quantitatively important pool of soil microbial biomass under C-replete, nutrient-limited conditions.
2. Microbial biomass growth is substantially underestimated by neglecting intracellular storage synthesis.
3. Due to low opportunity costs, surplus storage is likely to be quantitatively more important than reserve storage when measured across an entire soil community. Therefore, nutrient supplementation (N, P, K, and S) will suppress storage compound accumulation in favour of replicative growth.

Soil microcosms were incubated under controlled conditions, with C availability manipulated through additions of isotopically labelled ($^{13}$C and $^{14}$C) glucose, which is common in soil, including as a component of plant root exudates and the most abundant product of plant-derived organic matter decomposition[25]. A combined nutrient treatment (N, P, K, and S) provided inorganic fertilizers common in agriculture. A fully crossed design included three levels of C addition (zero-C, low-C and high-C; 0, 90 and 400 µg C/g soil) and two levels of nutrient supply (no-nutrient and nutrient-supplemented) with nutrients added at a level predicted to enable full C assimilation under the high-C treatment, based on microbial biomass C:N:P ratios typical of agricultural soil[26] and an assumed C-use efficiency of 50%. $CO_2$ efflux and its isotopic composition was monitored at regular intervals.

Microcosms were harvested after 24 and 96 h, with these incubation times selected to balance the synthesis of storage (previously observed over a timeframe of days[8]) with the risk of artefacts induced by recycling of labelled biomass[19]. Harvested soil was analysed for microbial biomass (by CFE), dissolved organic carbon (DOC), dissolved nitrogen (DN) and the storage compounds PHB and TAGs. In parallel, a set of smaller microcosms (0.5 g soil) was incubated under otherwise identical conditions to measure microbial growth as the incorporation of $^{18}$O from $H_2^{18}$O into DNA[20]. This method captures replicative growth better than tracing specific C substrates. Together these provide integrated observations of heterotrophic microbial biomass, growth and storage in a natural microbiome, examining the importance of storage as a resource-use strategy in response to environmental resource supply and changes in element stoichiometry.

## Results and discussion
### Microbial nutrient limitations and $CO_2$ efflux
We first describe observed patterns of soil respiration and dissolved nutrients that aid interpretation of the prevailing resource limitations during storage compound synthesis and degradation. Glucose addition stimulated large increases in $CO_2$ efflux (Fig. 1), primarily derived from glucose mineralization (Fig. 1B). Nutrient supplementation barely affected $CO_2$ efflux rates from the zero- or low-C additions and for none of the zero- or low-C treatments was N availability (measured as DN) significantly reduced relative to the control at 24 h (Supplementary Fig. S1A, B). Thus, C limitation dominated in the zero- and low-C treatments throughout the experimental period, irrespective of nutrient additions.

With high-C addition, $CO_2$ efflux rates under the two nutrient levels diverged strongly after 12 h, with the no-nutrient treatment declining steadily from 12 h until the end of the experiment. This early decline in mineralization was consistent with the onset of nutrient limitation, after microbial growth on the added glucose had depleted easily available soil nitrogen and driven up the C:N ratio of dissolved resources (Supplementary Fig. S1). This depletion in the high-C, no-nutrient treatment was reflected in suppressed DN after 24 h, with only 35.8–62.5% of the zero-C, no-nutrient control (family-wise 95% confidence interval; Supplementary Fig. S1). Nutrient limitation was accompanied by an accumulation of highly labelled DOC at 24 h in the soil solution, reflecting unused glucose or soluble by-products in an amount 19.6 ± 2.1% (mean ± standard deviation) of the original C addition (Supplementary Fig. S2). Therefore, high C addition without supplementary nutrients resulted in

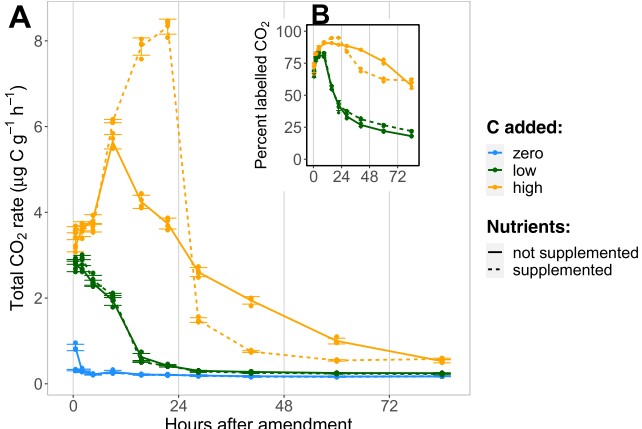

**Fig. 1 | Time-series of the $CO_2$ efflux from soil microcosms. A** Total $CO_2$ efflux following addition of a readily degradable $^{13}$C-labelled carbon source (glucose at 0, 90, and 400 µg C g$^{-1}$ soil) with or without mineral nutrient supply (N, P, K, S). Each point reflects the average rate of $CO_2$ efflux at the mid-point of the sampling interval. **B** Percent of total $CO_2$ derived from the added glucose. Error bars show standard deviation ($n$ = 4 independent soil microcosms).

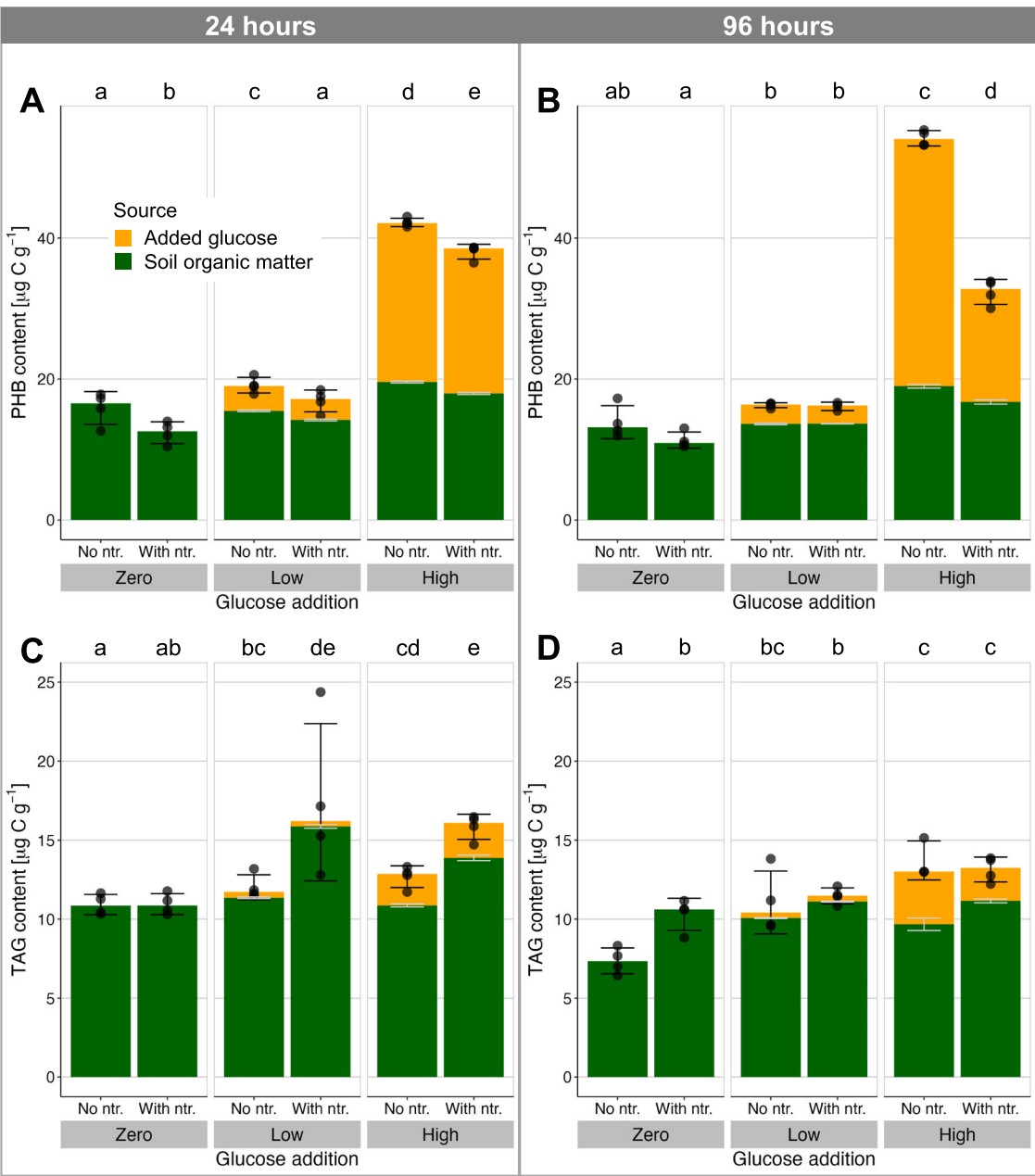

**Fig. 2 | Response of soil microbial storage to organic carbon and nutrient supply.** Storage compounds PHB **A**, **B** and TAGs **C**, **D** in soil 24 h **A**, **C**, and 96 h **B**, **D** after addition of a readily degradable, $^{13}$C-labelled carbon source (glucose at 0, 90 and 400 µg C g$^{-1}$ soil) with or without mineral nutrient supply (ntr.; N, P, K, S). The source of the stored C is shown in contrasting colours as determined by isotopic composition, with light grey error bars reflecting mean±standard deviation of the relative composition. Black error bars show mean ± standard deviation of the total storage compound pools, while colour bar heights show medians, as used in the robust analysis of medians ($n = 4$ independent soil microcosms, except for 1 treatment in each of TAGs and PHB where $n = 3$). Lowercase letters above the plots show post-hoc differences in total storage with $p < 0.05$ (2-sided pairwise comparison of medians with Benjamini-Hochberg adjustment for multiple comparisons).

rapid mineralization at first, but nutrient limitation set in within 12 h and continued for the remaining experimental period.

Nutrient addition had a strong effect in combination with high-C supply: it accelerated glucose mineralization until 24 h after addition (Fig. 1), after which CO$_2$ efflux dropped precipitously to below that of the high-C, no-nutrient treatment. For this high-C, nutrient-supplemented treatment, dissolved N decreased only moderately over 24 h (56.2–97.9% relative to the zero-C, no-nutrient treatment). With high-C addition after 24 h, DOC was far lower with nutrient supplementation than without (Cohen's d >> 1, family-wise $p < 0.001$), and DOC level for this treatment did not change further to 96 h, despite having higher N availability at 24 h than the no-nutrient treatment

(Cohen's d » 1, family-wise $p < 0.001$). This indicates that the microbial community had depleted the added C and re-entered C-limited conditions. Therefore, high C addition with supplementary nutrients maintained rapid C mineralization through the first 24 h, but glucose depletion then reasserted C-limitation for the rest of the experimental period.

**Presence and synthesis of microbial storage compounds**
PHB and TAGs were both found in the control soil (zero-C, no nutrients after 24 h; Fig. 2A, C), together representing a C pool 0.25 ± 0.03 (mean ± standard deviation) times as large as the extractable microbial biomass C (MBC, by CFE; Fig. 3). This ratio of stored C (PHB + TAG) to

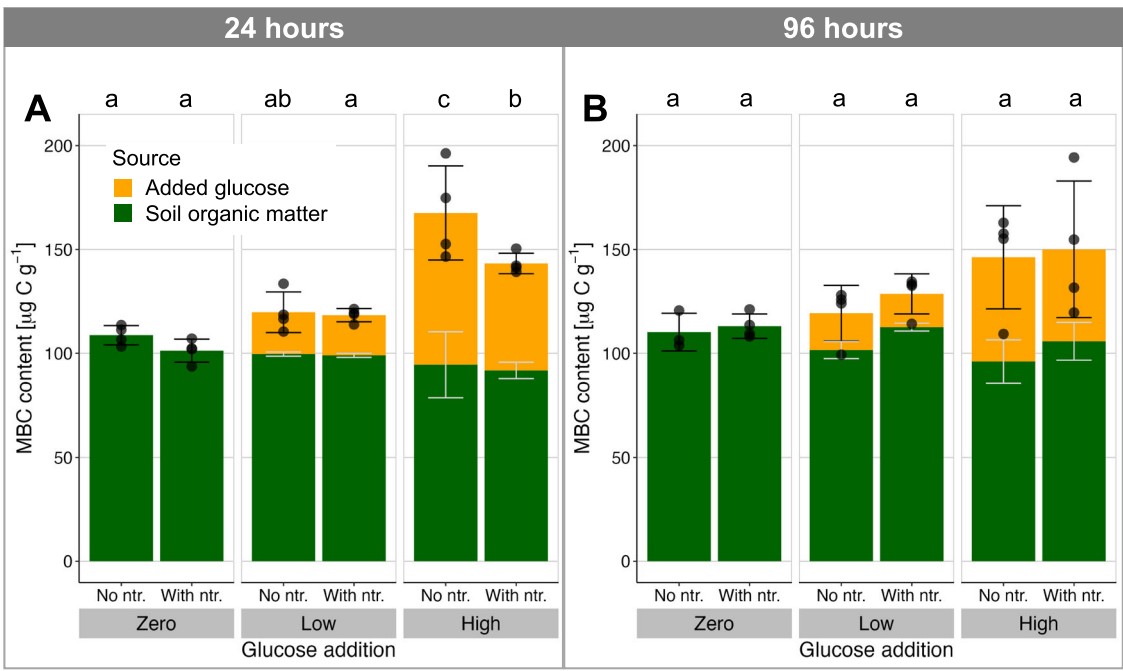

**Fig. 3 | Extractable soil microbial biomass determined by chloroform fumigation-extraction. A** 24 h and **B** 96 h after addition of a readily degradable, [13]C-labelled carbon source (glucose at 0, 90, and 400 µg C g⁻¹ soil) with or without mineral nutrient supply (ntr.; N, P, K, S). The heights of the bars represent the mean ±standard deviation as black errorbars (*n* = 4 independent soil microcosms except for one treatment with *n* = 3: zero glucose, no nutrients at 96 h). Contrasting colours reflect the source of the extractable biomass as determined by isotopic composition, with light grey error bars showing mean±standard deviation of the relative composition. Lowercase letters above the plots show post-hoc differences in mean total storage with *p* < 0.05 (2-sided Tukey HSD test, which adjusts for multiple comparisons). Corresponding C:N ratios are presented in Supplementary Fig. S3.

extractable MBC ranged from 0.19 ± 0.02 to 0.46 ± 0.08 over all treatments, indicating that storage is a significant pool of biomass not only under C-replete conditions, as hypothesized, but even when C availability is limited. Furthermore, the common measures of soil microbial biomass rely on extraction of water-soluble C after chloroform fumigation, which does not capture these highly hydrophobic storage compounds. This suggests that microbial biomass C may be widely underestimated in soil, and calls for methodological advancements to more systematically capture these (and possibly other) storage compounds in assessments of microbial growth.

The two storage compounds were both responsive to the supply of C and complementary nutrients (*p* < 0.01), but with very different behaviours. At both timepoints, the low input of C stimulated only a moderate increase in total PHB, irrespective of nutrient supply. In contrast, high C input stimulated a large increase in PHB, particularly when not supplemented with nutrients (a 308% increase over the zero-glucose, no-nutrient treatment at 96 h, with Hodges-Lehmann median difference of 36.0–42.9 µg C g⁻¹). In comparison, extractable biomass reflected a non-significant mean difference of only 33% between these treatments. Nutrient supply significantly suppressed PHB storage, even in the absence of added C (nutrient main effect, robust ANOVA of medians 24 h: $F_{(1,\infty)} = 35$, *p* < 0.001; 96 h: $F_{(1,\infty)} = 275$, *p* < 0.001). Isotopic composition ([13]C) indicated that assimilation of glucose C into new PHB continued between 24 and 96 h under the nutrient-limited conditions of the high-C, no-nutrient treatment (Hodges-Lehmann median difference of 10.2–13.3 µg C g⁻¹, 95% confidence interval), while extractable microbial biomass C showed no significant change. For the high-C, nutrient-supplemented treatment, the increased C limitation after 24 h induced degradation of PHB during this later incubation period (median reduction of 2.7–8.6 µg C g⁻¹). The PHB storage pool therefore responded dynamically to shifts in resource stoichiometry on a timescale of hours to days, with changes as expected from a surplus storage strategy. These observations are consistent with PHB biosynthesis in pure culture[27], which is stimulated by excess C

availability in diverse bacterial taxa[24]. This study demonstrates such microbial storage dynamics in a terrestrial ecosystem. At the end of the incubation, stored C across the various treatments was sufficient to support 109–347 h of microbial respiration at the $CO_2$ efflux rate of the zero-C, no-nutrient treatment (i.e., basal respiration). Much longer periods would be envisaged if accompanied by strong downregulation of energy use in response to the stress[28]. Storage could thus be a crucial resource for withstanding starvation or other stress. A surplus storage strategy is particularly effective at buffering microbial activity by levelling out fluctuations in resource availability and stoichiometry[11]. Furthermore, storage representing a substantial proportion of biomass offers a resource for regrowth following disturbance, indicating a potential role of storage in supporting resilience of this soil microbial community. In these ways, the resources stored in PHB could support the resistance and resilience of this soil microbial community against environmental disturbance[4].

Storage of TAGs was enhanced by C input (Fig. 2C, D), but its response to resource stoichiometry differed greatly from PHB. Over 24 h, nutrient supplementation stimulated more TAG accumulation, rather than suppressing it (main nutrient effect $F_{(1,\infty)} = 10.8$, *p* = 0.001 and nutrient:glucose interactions between zero-C and the two C-supplemented treatments, both *p* < 0.01), while over 96 h, nutrient supply had little effect with C addition and increased TAGs when C was not added (95% confidence interval for Hodges-Lehmann median difference 0.5–4.7 µg C g⁻¹). The TAG response to C and nutrient supply over 96 h resembled changes in extractable microbial biomass (Fig. 3), which was increased by C supply but not significantly enhanced by nutrients (ANOVA main effect of C supply at 96 h: $F_{(2,17)} = 7.1$, *p* = 0.006). Therefore, unlike PHB, TAG synthesis was not stimulated by a stoichiometric surplus of available C, suggesting a reserve storage function for this compound. Notably, the relative allocation of glucose C between PHB and TAG remained relatively constant (PHB:TAG ratio of glucose-derived C ranged between 7.0 and 11.5 across all treatments) because the C source used for TAG biosynthesis varied more

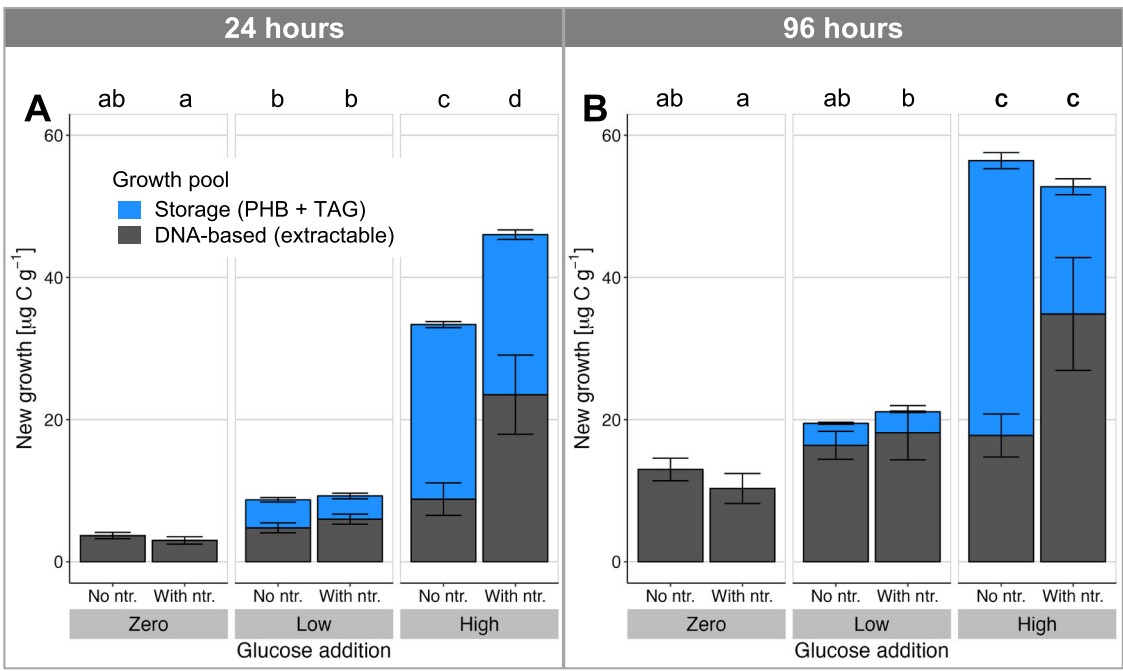

**Fig. 4 | Comparison of new storage biosynthesis with DNA-based microbial growth reveals storage as a substantial, overlooked component of biomass growth in soil.** [13]C-labelled storage compound synthesis (PHB and TAGs) and DNA-based growth (incorporation of [18]O) were measured in soil 24 **A** and 96 h **B** after addition of a readily degradable, [13]C-labelled carbon source (glucose at 0, 90, and 400 µg C g[-1] soil) with or without mineral nutrient supply (ntr.; N, P, K, S). Error bars represent mean±standard deviation in each component of the stacked bar (n = 4 independent soil microcosms). Lowercase letters above the plots show post-hoc differences in total observed growth with p < 0.05 (2-sided Tukey HSD test, which adjusts for multiple comparisons).

strongly than total TAG levels in response to C supply. This corroborates a reserve storage function of TAG, with total storage synthesis regulated independently of C supply and drawing on whichever C resources are available, whether glucose- or soil-derived. One advantage of a reserve storage strategy is that strategic stores are assembled even under conditions of chronic resource shortage. This allows for bursts of activity to support, for example, reproduction or transition to a resilient starvation state[4]. Therefore, while reserve storage may be quantitatively smaller than surplus storage (reflected here in the lower amounts and changes in TAG relative to PHB; Fig. 2), it can help communities to persist under conditions of sustained stress, and even exhibit resilience against additional disturbances.

A reserve storage function for TAG contrasts with most observations of TAG accumulation in pure culture in response to excess C[29]. Our observations also contrast with an earlier report that fungal TAG accumulation in a forest soil was largely eliminated by complementary nutrient supply[9], but much larger amounts of C were provided in that experiment (16 mg glucose-C g[-1]). The observed patterns of TAG storage are however consistent with abundant evidence of reserve storage among microorganisms, in particular that C storage occurs despite or in response to declining or limiting C availability[4]. For example, *Rhodococcus opacus* accumulated 21% of cell dry weight as TAG in the presence of excess N[30]. In our experiment C was traced into both bacterial (16:1ω7) and fungal (18:2ω6) TAGs (Supplementary material Figs. S4 and S5). The fungal biomarker 18:2ω6 was only a minor contributor to TAG incorporation in the current experiment, yet even this fungal TAG was not suppressed by nutrient addition. Our results suggest that both fungi and bacteria employed TAGs as a reserve storage form, with overall levels of TAG storage more closely linked to replicative growth than to resource stoichiometry.

In summary, the response of PHB storage to different C and nutrient conditions was largely consistent with the hypothesized surplus storage mode. In contrast, patterns of TAG storage were better characterized by the reserve storage mode. There is no a priori reason to expect distinct storage strategies to correspond to different

compounds, since both PHB and TAG can in principle provide C storage and mobilization under comparable conditions. Since some bacterial taxa can synthesize both PHB and TAGs[31,32], the question arises whether these compounds fulfil different storage functions in individual organisms, or whether the different responses emerge at a community scale, with each compound used by a different set of microbial taxa following divergent storage strategies. The first possibility would suggest as-yet unidentified differences in the metabolism of these compounds that distinguish them for different storage purposes. On the other hand, if storage strategy and preferred storage forms are correlated across taxa, then storage traits could prove useful as proxies of resource allocation strategy in microbial trait-based frameworks.

### Microbial storage as a component of biomass growth

The incorporation of C into soil microbial biomass is an essential step in the terrestrial C cycle[1], and appropriate estimates of these flows are required for understanding and managing ecosystem C balances[33]. We simultaneously performed a parallel experiment using identical treatments and temperature and moisture conditions to measure microbial growth using [18]O incorporation into DNA[20]. This method is calibrated to units of C based on extractable biomass from the CFE method, and therefore does not capture hydrophobic PHB or TAG storage. We compared the [18]O-DNA-based measure of growth with the incorporation of isotopically labelled glucose C into storage compounds (Fig. 4). This provides a comparison of magnitude using a lower bound for storage synthesis by neglecting the biosynthesis of storage from other C sources and any degradation of labelled storage during the incubation. Furthermore, only two storage forms were measured here, whereas other microbial storage compounds are also known[4]. Storage comprised up to 279 ± 72% more biomass growth than observed by the DNA-based method (for the high-C, no-nutrient treatment at 24 h, Fig. 4A). Even under conditions of C limitation (zero and low-C treatments), biomass growth through allocation to storage represented an additional 16–96% incorporation of C into biomass. Intracellular storage evidently plays a quantitatively significant role in microbial

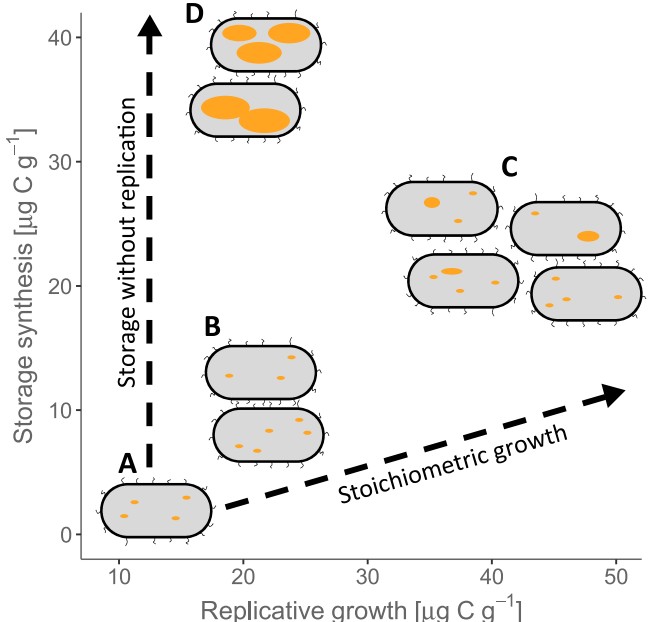

**Fig. 5 | Intracellular storage represents an alternative pathway for growth of microbial biomass.** In this conceptual figure the y-coordinates reflect the measured incorporation of added C into storage after 96 h, and the x-axis represents replicative growth measured by $^{18}O$ incorporation into DNA (see also Fig. 4). According to contemporary assumptions, all growth should follow the stoichiometric growth curve that maintains constant element ratios in the biomass (dashed line to the right). The microbial population is shown schematically by bacterial cells, with yellow lipid inclusion bodies representing storage. Without C supply, only low levels of replicative growth occur **A**. Low C additions (with ample nutrients) stimulate replicative growth and limited C incorporation into storage **B**, with the ratio of new storage to non-storage biomass staying close to that predicted by assuming constant biomass stoichiometry. High C addition with complementary nutrients stimulates both strong replicative growth as well as disproportionately large storage synthesis **C**, moderately violating the stoichiometric assumption. However, nutrient limitation switches growth strongly towards storage **D**, incorporating C into biomass with little replicative growth, closer to the extreme case of pure storage without replication than the assumption of stoichiometric growth.

assimilation of C under a broad range of stoichiometric conditions, and biomass growth would be substantially underestimated by neglecting storage. Microbial growth is a central variable in microbially-explicit models of the C cycle[34], so the substantial scale of storage also encourages a reassessment of model inputs and interpretation of results wherever short-term measurements or dynamic changes are involved. The important model parameter of carbon-use efficiency is typically measured over 24-h periods[35], but over this timeframe we observed storage changes that constituted a substantial component of the microbial C balance. This suggests that more nuanced representations of microbial metabolism and C allocation may be required to accurately account for microbial C use.

Microbial biomass growth is frequently understood as synonymous with the replicative growth of microbial populations. However, the incorporation of C into storage compounds represents an alternative growth pathway (Fig. 5), which differs from replicative growth in crucial ways. Models of microbial growth typically assume that increases in biomass match the elemental stoichiometry of the total biomass (the assumption of stoichiometric homoeostasis[36]), and therefore implement overflow respiration of excess C under conditions of C surplus[37]. However, substantial incorporation of C into otherwise nutrient-free PHB and TAG clearly does not follow whole-organism stoichiometry. Growth in storage therefore increases total biomass in a stoichiometrically unbalanced manner. The short experimental timeframe here is representative of environmental resource pulse and

depletion processes, such as the arrival of a root tip in a particular soil volume or death and decay of a nearby organism. Storage provides stoichiometric buffering during such transient resource pulses, which is predicted to increase C and N retention over the longer term[11]. By enhancing the efficiency with which microbes incorporate transient resource pulses and supporting metabolic activity through periods of resource scarcity, storage can contribute to the survival of microbes facing stressful habitat changes. Resource availability in natural and agroecosystems changes over various time-scales, and we hypothesize that microbial storage may also be responsive to, for example, seasonal changes in belowground C inputs, supporting microbial activity through resource-poor winter periods or dry summers. Moreover, storage enables a diversification of resource-use strategies, reflected here in the contrasting responses of PHB and TAG. Ecosystem stability is promoted by diverse strategies within the community[38], suggesting that storage can contribute to resistance and resilience of microbial communities facing environmental disturbances.

These findings encourage greater recognition of storage synthesis and degradation as pathways of microbial biomass change, in addition to cellular replication. Accounting for microbial storage as a key eco-physiological strategy can enrich our understanding of microbial resource use and its contributions to biogeochemical cycles and ecosystem responses under global change.

## Methods
### Experimental design
Topsoil (0–25 cm) was collected in November 2017 from the Reinshof experimental farm near Göttingen, Germany (51°29'51.0" N, 9°55'59.0" E) following an oat crop. Five samples along a 50 m field transect were mixed to provide a single homogenized soil sample. The soil was a Haplic Luvisol, pH 5.4 (CaCl$_2$), $C_{org}$ 1.4%[39]. Soil was stored at 4 °C for one week prior to sieving (2 mm) and then distributed into airtight 100 mL microcosms in laboratory bottles with the equivalent of 25 g dry soil at 48% of water holding capacity (WHC). Four replicates were prepared for each treatment and sampling timepoint. Microcosms were placed in a climate-controlled room at 15 °C and preincubated for one week before adding treatment solutions.

Treatment solutions provided glucose as a C source (0, 90 or 400 µg C/g soil) in a fully crossed design with added nutrients or a no-nutrient control (combined (NH$_4$)$_2$SO$_4$ and KH$_2$PO$_4$, respectively 0.613 and 0.106 µmol g$^{-1}$ soil). Glucose levels were selected to probe the effects of C supply on storage, with additions above and below the magnitude of MBC having potentially contrasting effects on microbial growth[40]. Glucose treatments contained uniformly isotopically labelled glucose (3 at% $^{13}C$ and 0.19 kBq $^{14}C$ per microcosm, respectively from Sigma-Aldrich, Munich, Germany and from American Radiolabelled Chemicals, Saint Louis, U.S.A.). The $^{14}C$ label in the added glucose enabled rapid and accurate measurement of glucose-derived C in liquid extracts by scintillation counting, while $^{13}C$ was traced in all other pools. The same amount of nutrients was used in all nutrient-addition treatments, with this set to be sufficient for the complete utilisation of all C added in the high glucose treatment, assuming a C:N:P ratio of 38:5:1 for an agricultural microbial community[26] and a C-use efficiency of 50%[41]. Addition of the treatment solutions raised the soil moisture to 70% of WHC, after which the microcosms were sealed with air-tight butyl rubber septa and their headspace flushed with $CO_2$-free synthetic air. Headspace gas was sampled with a 30 mL gas syringe at regular intervals and collected in evacuated exetainers (Labco, Ceredigion, U.K.) for measurement by gas chromatography–isotope ratio mass spectrometry (GC-Box coupled via a Conflo III interface to a Delta plus XP mass spectrometer, all Thermo Fischer, Bremen, Germany). After gas sampling, the headspace in each microcosm was again flushed with $CO_2$-free air. Microcosms were harvested 24 and 96 h after application of the treatment solutions. The soil in each microcosm was thoroughly mixed by hand for 30 sec and subsampled for chemical analysis.

## Chemical analysis

Extractable microbial biomass was measured by CFE[14,42]. A total of two 5 g subsamples of moist soil were taken from each microcosm. One was immediately extracted by shaking in 20 mL of 0.05 M $K_2SO_4$ for 1 h at room temperature, then centrifuged and the supernatant filtered. The other was exposed to a chloroform-saturated atmosphere for 24 h, after which residual chloroform was removed by repeated evacuation and the fumigated soil was extracted in the same manner as the non-fumigated subsample. Extractable MBC was calculated as the difference in DOC between the fumigated and non-fumigated samples, measured on a Multi N/C 2100 S analyser (Analytik Jena, Jena, Germany). CFE biomass is reported here as extractable biomass, without conversion with uncertain extraction efficiencies. Glucose-derived MBC was similarly calculated from the difference in radioactivity ($^{14}$C) of the extracts as measured on a Hidex 300 SL scintillation counter (TDCR efficiency correction, Hidex, Turku, Finland) using Rotiszint Eco Plus scintillation cocktail (Carl Roth, Karlsruhe, Germany). DOC and DN were determined respectively as organic carbon and total nitrogen in the extracts of the unfumigated soil.

PHB was determined by Soxhlet extraction of 4 g freeze-dried soil into chloroform, followed by acid-catalysed transesterification in ethanol and GC-MS quantification of the resulting ethyl hydroxybutyrate on a 7890 A gas chromatograph (DB1-MS column, 100% dimethyl polysiloxane, 15 m long, inner diameter 0.25 mm, film thickness 0.25 μm), with helium (5.0) as the mobile phase at a flow rate of 1 mL min$^{-1}$, coupled to a 7000 A triple quadrupole mass spectrometer (all Agilent, Waldbronn, Germany)[8]. Injection volume was 1 μL at an inlet temperature of 270 °C and split ratio of 25:1. The GC temperature was: 42 °C isothermal for 7 min; ramped to 77 °C at 5 °C min$^{-1}$; then to 155 °C at 15 °C min$^{-1}$; held for 15 min; and then ramped to 200 °C at 10 °C min$^{-1}$. The transfer line temperature was 280 °C, with electron ionization at 70 eV. Quantification was based on ions at m/z 43, 60, and 87 for the ethyl 3-hydroxybutyrate analyte, and at m/z 57, 71, and 85 for the undecane internal standard. Identity and purity of peaks was confirmed by scan measurement across the range m/z 40 to 300. The same chromatographic conditions were used for determination of the PHB isotopic composition on a Thermo GC Isolink coupled with a Conflo IV interface to a MAT 253 isotope ratio mass spectrometer (all Thermo Fisher, Bremen, Germany), but with splitless injection. The measured isotopic compositions were corrected for C added in derivatization[43].

TAGs were quantified as neutral lipid fatty acids as follows:[7] Lipids were first extracted from 5 g frozen soil into a single-phase chloroform–methanol–water solution, purified by solvent extraction, and neutral lipids separated from more polar lipids on a silica solid-phase extraction column. Following removal of the solvent by evaporation, the purified TAGs were hydrolysed (0.5 M NaOH in MeOH, 10 min at 100 °C) and methylated (12.5 M BF$_3$ in MeOH, 15 min at 85 °C), followed by extraction into hexane, drying and redissolution in toluene. The resulting fatty acid methyl esters were quantified by GC-MS on a 7890 A gas chromatograph (DB-5 MS column, 5%-phenyl methylpolysiloxane, 30 m coupled to a DB1-MS 15 m long, both with an inner diameter 0.25 mm and film thickness 0.25 μm) with an injection volume of 1 μl into the splitless inlet heated to 270 °C, and at a constant flow of helium (4.6) of 1.2 mL min$^{-1}$, coupled to a 5977B series mass spectrometer (Agilent, Waldbronn, Germany), set to 70 eV electron impact energy, with the GC oven programme as follows: initial temperature 80 °C isothermal for 1 min, ramped at 10 °C min$^{-1}$ to 171 °C, ramped at 0.7 °C min$^{-1}$ to 196 °C, isothermal for 4 min, ramped at 0.5 °C min$^{-1}$ to 206 °C, and ramped at 10 °C min$^{-1}$ to the final temperature of 300 °C, isothermal for 10 min for column reconditioning. Isotopic composition was determined in triplicate using a Trace GC 2000 (CE Instruments ThermoQuest Italia, S.p.A), coupled with a Combustion Interface III to a DeltaPlus isotope-ratio mass spectrometer (Thermo Finnigan, Bremen, Germany) using the same GC parameters.

Growth was estimated by $^{18}$O incorporation into DNA[19,20]. Parallel microcosms were prepared with 0.50 g soil in 2 mL Eppendorf tubes (Eppendorf, Hamburg, Germany) and incubated alongside the larger microcosms. This smaller scale was necessitated by the cost of $^{18}$O-water. This is nevertheless larger than the soil amounts typically used for DNA extraction, which achieve consistent measures of bacterial and fungal community composition. This is also orders of magnitude larger than the scale of microbial interactions[44]. These considerations, alongside the care taken to ensure identical conditions of temperature, moisture and handling, give confidence that this incubation was representative of the same processes occurring in the larger microcosms. Treatment solutions were prepared at the same concentrations as for the larger microcosms, but enriched with 97 at% H$_2$$^{18}$O so that addition provided a final soil solution of 4.2 at% $^{18}$O. Tubes were withdrawn from incubation 24 h and 96 h after addition and immediately frozen at −80 °C. DNA was subsequently extracted using MP Bio FastDNA Spin Kit for Soil (MP Biomedicals, Solon, OH, USA). DNA concentration in the extract was measured on an Implen MP80 nanophotometer (Implen, Munich, Germany) at 260 nm, with A260/280 and A260/A230 to confirm quality, and 50 μL was pipetted into silver capsules, freeze dried, and measured by TC/EA (Thermo Finnigan, Bremen, Germany) coupled with a Conflo III interface to a Delta V Plus isotope ratio mass spectrometer (all Thermo Finnigan, Bremen, Germany). The total measured O content of the sample, the O content of the DNA (31% by mass), and the $^{18}$O natural abundance of unlabelled control samples were used to calculate the background $^{18}$O from the kit. This background $^{18}$O was deducted to obtain $^{18}$O abundance of the DNA, which was applied in a 2-pool mixing model with 70% of O in new DNA derived from water[45] (model detailed in Supplementary B). This provided the fraction of extracted DNA that had been newly synthesized during the incubation period. This fraction was multiplied by extractable microbial biomass to arrive at gross biomass growth in units of μg C g$^{-1}$ soil.

## Statistical analysis

Statistical analysis was performed in R[46] with preliminary calculations in Microsoft Excel (version 16.67). Results for $CO_2$, MBC, DOC, DN, TAG, PHB, and isotopic compositions were calculated for each independent sample and reported as mean ± standard deviation for each treatment group, unless otherwise noted. Comparisons between these pools were similarly calculated at the sample level before expressing as mean ± standard deviation.

DN and DOC data were log-transformed to satisfy assumptions for ANOVA (Shapiro-Wilk's test of normality and Levene's test for homogeneity of variance), followed by Tukey's HSD test for pairwise comparisons of treatment effects. The same analyses were performed on untransformed extractable microbial biomass data. Ranges for treatment effects on DN, DOC and MBC reported in the text reflect 95% family-wise confidence intervals from pair-wise Tukey's HSD tests. Where relevant, effect sizes were computed as Cohen's d, using the effsize package[47].

Levels of labelled storage compounds showed considerable heteroskedasticity that could not be consistently corrected by transformation, particularly due to very high levels of unsaturated fatty acids in one of the 24 h samples. This conceivably reflected a hotspot of fungal activity in the soil. This datapoint was therefore conservatively retained since this would comprise relevant variability in the soil. Analysis of storage compounds (PHB and TAG) proceeded by robust ANOVA of medians for each timepoint separately using the R package WRS2[48]. Consistent with the median-based robust ANOVA, storage differences between treatments reported in the text are median differences, with uncertainty given as 95% confidence intervals calculated by the Hodges-Lehmann estimator (R package DescTools[49]). Comprehensive pairwise post-hoc comparisons of medians was performed using medpb to provide significance indicators in figures (Fig. 2) (R package

WRS[50]), with Benjamini-Hochberg adjustment of p values for multiple comparisons.

Growth estimation by [18]O incorporation used DNA concentration and its [18]O enrichment to determine mean gross microbial growth for each treatment in relative terms, and the associated standard deviation. The corresponding mean extractable microbial biomass values were applied to convert to absolute units of μg C, using standard rules of error propagation[51], to provide the DNA-based measure of mean microbial biomass growth for each treatment. These DNA-based growth estimates were combined with the mean production of labelled storage compounds (sum of C in glucose-derived PHB and TAG), again using rules of error propagation, to obtain estimates of total (DNA-based and storage) mean biomass growth and associated standard deviations. These were subjected to 2-way ANOVA and Tukey HSD to test the significance and size of treatment effects (Fig. 3). Arithmetic comparisons between MBC, growth and storage pools (for example, the relative scales of DNA-based growth and storage growth) were calculated using mean values with error propagation.

### Reporting summary
Further information on research design is available in the Nature Portfolio Reporting Summary linked to this article.

## Data availability
All source data generated in this study has been deposited in the Zenodo open data repository[52] under https://doi.org/10.5281/zenodo.6386047. This data is publicly available.

## Code availability
R scripts used for data analysis are publicly available on the Zenodo open data repository under https://doi.org/10.5281/zenodo.6386047.

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

## Acknowledgements

We gratefully acknowledge the laboratory assistance provided by Kali Middleby, Andrew Gall, Lydia Köbele and Karin Schmidt. We gratefully acknowledge the financial support of the Deutscher Akademischer Austauschdienst (DAAD; 91525994) for a fellowship supporting the experimental work (K.M.J.), the Dutch Research Council (N.W.O.) for funding of the Veni project VI.Veni.202.086 (K.M.J.), and the Open Access Publication Fund of the University of Tübingen (M.A.D.). This work is associated to the DFG Priority Program 2322 SoilSystems, project EcoEnergetics (DFG DI 2136/17–1; M.A.D.). We thank the staff of the core projects of the SPP and the scientific committee for establishing the SPP project.

## Author contributions

K.M.J., A.B., C.C.B., and M.A.D. jointly initiated and designed the experiment; K.M.J., A.B., and J.D. conducted the experiment and analyses; K.M.J. and A.B. undertook data analysis; K.M.J. wrote the manuscript with assistance and comments of all co-authors.

## Funding

## Competing interests

The authors declare no competing interests.
