## [Peer Review File · Nature Communications]

Intracellular carbon storage by microorganisms is an overlooked pathway of biomass growthREVIEWER COMMENTS

Reviewer #1 (Remarks to the Author):

Comments:

The manuscript "Intracellular carbon storage by microorganisms is an overlooked pathway of biomass growth" by Mason-Jones et al. investigates the relevance and build-up of microbial storage compounds. They argue that specifically PHB and TAG are overlooked storage compounds in soil microbial communities that are currently overlooked. The production of storage compounds also constitutes microbial growth that is not captured by state-of-the-art methods.

I think this is a very relevant and timely study, that opens up new lines of thinking and unexplored avenues in soil biogeochemistry. As such I think the study is of great interest to the whole community and to the broad audience of Nature Communications. I strongly encourage its publication after the following issues are addressed:

1) How much of the added ^{13}C was recovered in the CFE extracts? Was this proportional to DNA-growth or could this constitute another, water-soluble, pool of storage compounds? Also was there a correction for extraction efficiency done for the CFE extracts?

2) To calculate C-growth with the ^{18}O method, increases in DNA are converted to units of C using the CFE derived MBC. How does DNA increase alone relate to the production of storage compounds? Also, could storage compounds be considered in the calculation to convert DNA production into units of C?

Minor comments:

Line 58: replace "predicts" with "indicates"

Hypotheses: Hypothesis 3 is in my opinion directly related to hypothesis 1. I think they should be right after one another or could even be combined

Line 82: delete: "the turnover of...captures"

Line 86: change to: "supply and changes in element stoichiometry"

Line 129: change to " $24.7 \pm 2.5\%$ (mean \pm standard deviation) of the extractable microbial biomass"

Figure 2: Also add the letters that indicate the respective panel to (left) and (right)

Line 305: You should mention here if you used a factor to account for CFE extraction efficiency.

Reviewer #2 (Remarks to the Author):

Nature Comms NCOMMS-22-23016

Review

This is a review of the paper "Intracellular carbon storage by microorganisms is an overlooked pathway of biomass growth." This paper is a major advance to our considerations of the impact of microorganisms on the carbon cycle because it sheds light onto microbial-derived compounds (i.e., triacylglycerides (TAG) and polyhydroxybutyrate (PHB)) currently missed or ignored, specifically compounds used to temporarily store carbon for later use in macromolecular synthesis. They find that the two storage compounds represent substantial portion of microbial biomass growth following the addition of labeled glucose substrate and that these responses differ under differing nutrient conditions. I found this paper interesting and thought the findings would be a useful contribution to our understanding of soil carbon storage and microbial-derived carbon sequestration – a very timely topic right now. Despite my interest in this important and timely topic, I have several major concerns about the manuscript that I believe should be addressed prior to publication. These include: the rationale of the study as important to ESMs, shortcomings of the figures, background on the extraction methods and storage compounds, and gaps in the methodology or explanation of the methodology.

Rationale of the study as important to ESMs: While the authors discuss this finding as a major contribution to considering carbon use efficiency (CUE) and microbial biomass in Earth system process

models, I think their more major impact is on considering how these storage compounds contribute to the stability, resistance, and resilience of microbial communities. My rationale for this is not that refining microbial CUE is unimportant, but rather that CUE in many process models is too coarse of a measure (it's not dynamic and/or emergent, but rather static and/or a fixed parameter) to be influenced by this new transitory pool. Because the authors find that these storage compounds are a transitory pool (almost like an intermediate in a chemical reaction), adding them into process models that operate at global scales might not have much impact. Further, the authors suggest that these compounds should be considered part of the active biomass pool in ESMs. I also disagree on this point because in microbial-explicit models where the size of the biomass is consequential to process rates, biomass should be a proxy for the active number of cells. As such, whether there are many or few storage compounds should probably not be consequential to the active biomass pools nor should it affect process rates as reflected in ESMs. What the authors should perhaps be suggesting is that microbial metabolism and the possible fates of metabolism are more accurately accounted for in models. I think the authors could suggest that these storage compounds be incorporated to process models that operate at a finer level of organization (e.g., community, or perhaps even ecosystem) and those that require a more accurate accounting of the resistance and resilience of communities and total biomass, but that this is unlikely to be accurately represented in models that operate at the Earth system scale.

Despite my druthers about the reliance on the argument that this is an important finding for ESMs, I do agree that refining concepts of microbial CUE are important. The "conventional" idea that all carbon goes to either growth or respiration is an oversimplification. I'd suggest the authors consider a paper following on this more data-driven paper suggesting a refinement of CUE with these storage compounds, but also considering other non-biomass microbial C pools (like enzymes, antimicrobial compounds, secondary metabolites). Moving forward with this paper without the reliance on claiming that the importance of this work is relevant to ESMs, the authors will need an alternate argument about the importance of their work. I'd suggest a larger reliance and deepening of the arguments about the importance of these storage compounds for the stability of microbial communities, like that currently discussed in lines 199-203. This was fascinating. I'd like to see more separation of the concepts of the two storage strategies (i.e., surplus and reserve storage) and their impact on stability, resistance and resilience. Perhaps of service to this point would be a deeper contextualization in the discussion about what is known about the genetic regulation of TAG and PHB storage and if this matches the authors findings about the storage strategies associated with each, and with the authors questions about whether or not storage mechanism regulation differs among taxa or is relatively similar across all taxa in a community.

Figure shortcomings: The manuscript will also be aided by amending a couple of figures. When talking about nutrient limitations, showing C:N ratios rather than (or in addition to) DOC and TDN would be useful. For example, the point the authors are making in line 102 would be aided by a comparison of C:N in biomass and SOM, or at least a figure of soil nutrients. Also, figure S3 does not support the statement in line 113 and in fact seems to be contradictory. The error on the green and yellow figures is also unclear. Is the error for both bars? Or is there other shading I can't see? Error bars (as used on other stacked bars) are easier to interpret. Lastly, I'd really like to see the posthoc differences on the figures. It's hard to know where the statistical differences are by looking at the figures.

Background on the extraction methods and storage compounds: Further, I feel the introduction lacks a bit of context and should be refined to include (1) information about what different extractions methods do and don't extract/show and (2) information about the storage compounds. I suggest that text explaining what is observed with conventional biomass extraction methods vs. these TAG and PHB methods would help to support lines like 153 (that this is the first study to show microbial storage dynamics in a terrestrial ecosystem). Furthermore, the observations of shifting C:N in biomass relative to soil/solution C:N seem consistent with previous observations of biomass responses to changing nutrient conditions (i.e., C storage during nutrient scarcity (increase in C:N), turnover during nutrient abundance). However, the authors claim that the previous measures wouldn't include these storage

compounds. I was left wishing for more detail on what exactly is extracted by each type of extraction so I could understand whether this was an appropriate parallel or not. (2) And relatedly, it's important that the authors tell us more about the structure and solubility of these compounds, as well as where (in, on, or around the cell) these compounds are stored in order for the reader to fully get the whole picture and understand how we can structure this new information within the scaffold of what we know (like extraction of biomass into polar solvents like water and K₂SO₄). PHB and TAG should be explained as individual compound (classes?) in order to provide rationale for keeping their results separate in figure 2.

Gaps in the methodology or explanation of the methodology: 1) I am concerned about the comparability of the two incubation experiments given the large difference in biomass between the 180 incubations and the primary incubations. Ecological theory (species area relationships) predicts that the reduction in incubation size may lead to greater stochasticity in the diversity and community composition of the smaller microcosms and it's unclear how these differences are likely to impact the biomass and growth rate results they observe, therefore, the authors should provide evidence that the reduction in size of the 180 microcosms did not substantially influence their results between the two incubations. 2) I found it difficult to understand exactly how the authors calculated various percentages and ranges of carbon contributions. I think this could easily be clarified through more detailed methods (potentially supplemental if space is limiting). Ideally the authors could provide code supporting their calculations, but at the very least they should provide the equations used in these calculations (more details are in the line edits below).

I recognize this has become quite a large list, but I do hope it helps in the revision of this important work. Many of the line edits below are redundant to my overview points above, but some are new.

Best,
Dr. Jessica Ernakovich

Line edits:

1 – suggest that the title be changed to “Non-replicative carbon storage by microorganisms is an overlooked pathway of biomass growth”

25-26 – these concepts really come out of nowhere. While I really liked these in the manuscript and am suggesting even heavier reliance on them overall, they need more build up in the abstract

38 – As mentioned in the overview, more information on the structure and location of these compounds is needed. While there are likely many places in the intro that this can go, I like after line 38 could be good.

42-51 – This paragraph would be a great place to include more context about what compounds are extracted with the biomass methods. Line 45-46 about the DNA-based methods should be more direct.

48-51 – consider making this the topic sentence

53-60 – More details on reserve storage would be helpful, as it's less intuitive than surplus storage. For example, what do the culture studies find? Do we know this from one microbe, or are there culture studies across a wide range of bacteria? Also, it's unclear from the hypotheses whether this study can and will be able to unravel these. Are the experiments truly set up to test this?

Line 63-64 – This hypothesis is wishy-washy and weak. What is meant by “a substantial portion?” How can this be tested?

Line 66 – Replace “complementary” with more specific language

Line 68 – I challenge the concept that these are biomass, particularly in the context of this study showing that these compounds are a transient pool.

73 – this phrasing implied to me that these nutrients were tested separately rather than in tandem. Please amend.

76 – predicted by what? C:N? Please include assumptions.

77-79 – where these measurements isotopically-enabled? I presume yes, but please include.

Results and discussion – please flesh out surplus vs reserve storage concepts and how this work addresses these mechanisms.

95 – it is unclear what “these” is referring to

105 – I don’t see a decline in TDN between high C no nutrient conditions between 24 and 96 h in this figure. This should be refined or clarified. Does this conflict with line 112? I am finding this confusing to follow.

113 – I recommend being very specific about which control is being referred to here, as it is confusing to interpret the figures. It appears this comparison refers only to the no-nutrient addition incubation, but that doesn’t align with the graph. When I look at Figure S3, I see that the DOC without the N+P is greater than with N+P. But this line says the opposite (I think, but the wording is confusing). This needs to be fact checked and/or re-worded. More generally, I think the authors should include the calculations when appropriate in the methods (or an extended supplemental methods) section. This would greatly clarify for the reader exactly what is being compared to what.

130 – what is the ratio being referred to? I think you have added together the PHB and TAGs and then taken the ratio of that value and the MCB ug C/g here, but it is not entirely clear from the text. I recommend specifying these and other calculations in either the methods, or an extended supplemental methods section.

139 – include a p-value

142 – to be consistent, use the term zero-C rather than control

154-156 – it is unclear where the data is to support this assumption. Please provide a reference or details of how this basal respiration period was calculated.

180 – For the PHB:TAG comparison please specify units in text (is it in % of total ug C/g or a simple ratio of ug C/g of each storage compound?).

Paragraph beginning at line 195 – I would find it helpful and more appealing to a wider audience, if there was a discussion about what is known about the genetic regulation of PHB and TAG genes in pure culture or other systems, and if these patterns of regulation are consistent with the data presented here. For example, does the genetic regulation of PHB or TAG suggest that they are used for different storage purposes? Does regulation of these two gene cascades seem to be similar or vary a lot among organisms? (Note: I’m not suggesting the authors carry out a separate genomic analysis, but mining the literature for this information would lead to a richer discussion of the implications of these findings).

198 – clarify utilize... synthesize? Degrade? typo in the spelling of “fulfill”

199-203 – this is fascinating. I'd love this to be more central.

Section 2.3 – Given that the 180 experiments were performed on significantly less soil than the storage incubation, I'm concerned that the two incubations are not comparable. Even for well-homogenized samples, by random chance and from ecological theory (species area relationships), we expect that smaller incubation should have large differences in diversity and turnover between replicates simply due to ecological drift, therefore, I think it is important for the authors provided evidence that the small incubation are a good proxy biomass and growth rate for the larger incubations despite these differences in community structure.

211 – add "DNA-based" in front of 180 to add clarity

207 & 222– I fail to be convinced that these storage compounds should be considered biomass rather than microbial products because in models and in reality, biomass pool size regulates C turnover due to its action on other C compounds. These storage compounds have no agency, so until they are incorporated into structures that can act on SOM, I argue they should not be considered biomass.

218 – rather than C limitation, do the authors mean in the low C addition? Or do you know where a threshold of C limitation is? Even with no C added, there isn't necessarily a C limitation. I'd argue that C limitation (here or in the intro) need to be defined and that the level at it occurs in this specific soil should be discussed.

Line 222 – I appreciate the authors caveat here that this is likely to be most significant for short-term dynamics.

236 (and as discussed elsewhere) – Given that this data shows this is transient, I take issue with this being called a growth pathway. Instead, perhaps it should be considered a carbon allocation strategy.

Figures –

Figure 1 – a complementary figure showing the % CO₂ that was labeled (rather than the current inset) would help so the reader doesn't have to eyeball this. Also, please show the x axis (for both the main and the labeled) in 12 hour increments to match the text and align with the harvests. Also, the y-axis in the main panel should be labeled "Total CO₂ rate" to distinguish it from the "Labelled CO₂ rate" inset. It was unclear to me originally that the main figure represented both labelled and unlabelled fractions.

Figure 2 – the x axis labels should be less redundant so people only looking at figures can understand them. For example, both the glucose and nutrient treatments include a "none" to explain them. Also, since the nutrient addition wasn't just N+P, maybe this should be coded differently. Also, the error (shown as shading) is unclear. Is the shading just for the green or also the yellow? I'd suggest doing the error bars as in figure 3 or separating the components into separate figures. Additionally, this figure might be improved by adding an indicator (perhaps a star or dashed horizontal line) of TAG and PHB levels to the chart. This would help clarify the discussion of the ratios of stored carbon to biomass carbon on line 128. I'd also suggest showing the post-hoc differences on the figure to make this more informative.

Figure 3 – I suggest adding a top panel here with the chloroform fumigation data (from S2) because I needed to do some eyeballing to support line 176.

Supp Figure 1 – state that there is no statistical difference or show that. Also, showing the C:N ratio would help support the point in lines 97-98. Also, the caption needs more details (and punctuation).

It's not clear from the caption what the figure describes.

Supp Figure 2 – change carbon to glucose

Supp Figure 4-6 – These figures are inconsistent with the formatting on the other figures. I would appreciate it if these were brought into alignment, but at a minimum the authors should define what the colors represent in the captions. I assume Glc refers to glucose, and that GZ, GL, and GH are representing the different glucose treatments. But this was not easily or immediately clear. In addition, it's unclear which compounds are important on these figures. It would also be helpful, if the fungal and bacterial biomarkers referred to in the text were indicated on the x-axis of the graphs in S4 and S5, as the x-axis names differ slightly from those used in the text. Perhaps detail can be added to the caption.

Supp Figure 6 – Are these glucose-derived? Why a new figure format?

Methods

Line 274 – How long was the transect?

Line 276 – How long was the soil stored at 4 degrees?

283 – please include nutrient concentrations

284 – it is unclear whether ^{13}C and ^{14}C were added together and why

286 – I'm a little confused about how this ratio was determined for the 0 carbon added treatment.

More details about the stoichiometric ratios of C:N:P in each treatment would be helpful. Was the same concentration of nutrients added in the low C treatment as in the high carbon treatment? Please clarify in text.

294 – I'm unclear about the reasoning behind the 24h vs 96h time choices, it would be nice if the authors included some insight into this choice.

Line 310 and Line 326 – Please add to the text the number of grams of soil used for the PHB and TAG analyses.

327 – 328. End sentence with "in soil.³⁸" and start new sentence with "Lipids were first..."

Line 345 – I haven't done this kind of incubation or this small of an incubation, but I'm not sure that an incubation of 0.5g of soil in 2mL tubes is comparable to the 100mL microcosms containing 25g of soils. Would there be enough room for fungal proliferation/spatial distributions that can occur in the larger tubes to occur in the smaller? Additionally, would they not dry out during the pre-incubation? Were they also subject to the pre-incubation? Generally I'd just like more details here and some proof that the 2ml Eppendorfs equivalent to the larger microcosms

359 – please show mixing models and a definition of end members

Reviewer #3 (Remarks to the Author):

The purpose of this study was to quantify whether microbial storage compounds – specifically PHB and TAG – could comprise a significant portion of microbial biomass, and thus represent an important stock of microbial biomass C that is unaccounted for in traditional estimates of biomass growth, like chloroform fumigation extraction or DNA-based methods. The authors also measure the storage compounds PHG and TAG under different C and nutrient conditions, and compare them to extractable C and growth via the ^{18}O - H_2O method.

The authors present novel results that provide some of the first quantitative estimates of the relative importance of storage compounds in microbial biomass estimates. Based on the range of values they show, they make a convincing case that it is important to consider storage as a pathway for biomass growth, and that our current estimates likely underestimate actual biomass/growth values, especially under certain C and nutrient scenarios.

However, there were a few key issues with the manuscript. First, the authors do not very clearly spell out the broader implications of accounting for storage compounds in soil microbial growth/biomass. While this may be very clear to soil microbial ecologists/soil biogeochemists, it may not be to a broader readership. This could be expanded on both in the Introduction and much more thoroughly in the Discussion. Second, the entire first section of the main text (Section 2.1) does not feel very relevant to the point of the paper, which is about the role of storage compounds. If the authors were to keep that section, I would suggest moving into supplementary, as well as Figure 1. As it stands, it is confusing why the authors jump into a discussion on CO₂ efflux under different nutrient conditions on a paper whose main take home is supposed to be about storage compounds. This brings up the third point – the paper felt a bit thin on data, even if some interesting results are presented about the role of TAG and PHG. At core, its really two main data figures (Fig 2 and 3), especially if Figure 1 was moved to Supplementary. Fourth, it is unclear why the incubation was relatively short and why the particular storage compounds TAG and PHG were exclusively measured. There may be good reasons for both, it was just not clearly spelled out to me. Last, I found the final figure to be confusing. I think this presents a good opportunity to clearly spell out terms used in the paper, differentiate different hypotheses, and show results, and I think I could do that more effectively.

I detail some suggestions further in the line edits below:

Abstract:

Overall, I think you need to make a stronger case here for the broader applicability of the findings in your paper. It doesn't seem the main selling point should be that it helps explain the "mechanisms underlying resistance and resilience of microbial communities" (i.e., last line of the Abstract) as the paper doesn't really focus on this, but rather should focus on how it is essential to our accurate understanding of soil carbon cycling.

21: What do you mean they accounted for 20-46% of extractable biomass? I thought these storage compounds were not extracted in the chloroform fumigation method? And in line 130, you state that storage C is 20-46% as large as the extractable biomass pool? Maybe you mean the same thing here, but it is confusing to me. I would state the size of this pool of storage C relative to other estimates.

Introduction:

30: This opening sentence feels like its lacking something – specifically, why is this important? Isn't it important because it determines the flow of C and other nutrients through these systems?

37: Why highlight PHB and TAG? Are these the most quantitatively significant? The ones we are best at isolating? The most widespread? It feels a little early in the paper to give such specifics, especially with no context for why you are targeting these two specific compounds. I would perhaps save this for a later paragraph, and give me context as what these storage compounds are, and why you are focusing on them.

41: I still think this needs a stronger sell for why this is a major oversight that needs correcting. What is inaccurate in our current estimates of microbial processes and C cycling that would change by accounting for storage compounds?

49: What do you mean by 'local ecological stoichiometry'? I find this confusing. Can you more clearly state here examples of where biomass growth is important to know about? I definitely think microbial-explicit models is one good example.

51: Any good reference to cite here?

52: I would add "interpretation of microbial storage patterns"

53: I would italicize 'reserve storage' and 'surplus storage' or put in quotes, the first time you use them. It could also be very useful to show a simple conceptual figure here of definitions and predictions, as your Figure 1.

66: "At the community-scale ..." Not sure what this means. Compared to what? Individual? Population? Or compared to reserve storage? Not sure what you are comparing here.

68: How are hypothesis 3 and hypothesis 1 different from one another? Is it because #3 describes growth, whereas #1 is just about biomass as a standing stock? They seem very similar to me.

71: Why did you use both ¹³C and ¹⁴C labels? Briefly describe here, and in more detail in the main text. This seems like a key point in your study design, and in what makes it unique, so it would be useful to provide more explanation.

72: Glucose is also common in dissolved organic carbon from root and shoot plant litter.

75: What are these amounts based on, in terms of high and low? Reference? Biological scenario?

78: Why do such a short incubation? Please explain the justification here for a 96 hour incubation.

85: It seems a bit premature to say this study reveals the importance of storage in a natural microbiome, without yet stating how important it is!

Results

Section 2.1

I think this first section should directly followed from the first hypotheses. The first thing I'd like to see is therefore how much microbial storage compounds account for, in terms of total biomass (i.e. the first hypothesis). This should form the first section and Figure 1, as it is the most critical part of the paper. It is confusing that you begin the first sentence of the results by talking about patterns of soil respiration, and how they align with past observations. How is this relevant to the main point of the paper?

Overall, I'm confused how the first three paragraphs tie to the main purpose of this section? Perhaps movethese to supplementary, or put later on in the paper? I'm expecting to hear about storage compounds right off the bat, but this section is about CO₂ efflux rates under different nutrient levels. If the point of this is to show your treatments worked, again, I would put this in supplementary, including your current Figure 1.

Section 2.2

It would be useful here to briefly remind reader how you used isotopes to parse apart different contributions of storage compounds.

129: I found the wording "a pool 25% as large as" to be a confusing turn of phrase. Do you mean it was 25% the size of the extractable biomass pool?

132: "Storage equivalent to a substantial proportion of biomass" Not sure what this means?

137: Instead of "widely underestimated" I would just state the numbers ... by about 20 to 45%.

Section 2.3

207: Insert some references here?

218: "storage growth" feels like a confusing term. Maybe "allocation to storage compounds" or something along those lines?

223: I like this, as it starts to tangibly explain the implications of this study – can you include a bit more detail on how it would necessitate a reassessment of model inputs?

234: This is a great, clear paragraph, and would be very useful if it came earlier on – such as in the Introduction!

242: I appreciate the justification here of conducting such a short incubation, but it would be great if this came earlier. And how might these processes differ over longer time periods than 96 hours, such as sustained conditions of C or N limitation (or other relevant conditions) over the course of a growing season?

251: I'm finding Figure 4 a bit confusing, and it feels like its requiring too much time reading a long caption to understand what the figure is supposed to represent. A few thoughts: why does it require the particular X and Y axis? Could it instead directly include the stoichiometric conditions on the figure itself, so the reader doesn't need to refer to the caption? Why are the pie charts necessary? They all include the same ratio of PHB and TAG, so this feels confusing at first, as I'm looking to see if the ratios are different. Its not intuitive to me at first that different sized pie charts indicate different amounts of storage compounds. I think the yellow circles inside the cell do a much better job of this. Are the terms "storage growth" and "stoichiometric growth" used elsewhere in the paper? It seems not. I would use these earlier on, clearly define them, and use them throughout, so its very clear to the reader what they are, when they are being used in the final synthesis figure. It could also be useful to contrast the two forms of growth in a conceptual figure like this, by showing equal amounts of growth, but in one instance there are more individuals, and in another there are fewer, but more storage compounds. In this figure, it seems there always the same number of individuals (3).

267-270: I think you need a longer, more in depth discussion of the implications of this work. It could also be useful to discuss some of the limitations of this study, and key next steps.

Detailed responses to reviewer comments on the manuscript “Intracellular carbon storage by microorganisms is an overlooked pathway of biomass growth”

16 December 2022

Reviewer 1

1. How much of the added ^{13}C was recovered in the CFE extracts? Was this proportional to DNA-growth or could this constitute another, water-soluble, pool of storage compounds?

Recovery of glucose-derived C in MBC is now provided in Fig 3. It roughly tracks DNA growth, but there are some pitfalls to interpreting this too deeply: CFE reflects labelled compounds in the cytosol even if these have not yet been metabolised, and especially at the 24 h timepoint this may contribute strongly to the ^{13}C signal. Also, it is expected that the amount of labelled substrate addition would be reflected in the recovery in extractable biomass, but since the label was added as glucose, label addition also correlates with growth. In this case it is not easy to separate the effects of ^{13}C label addition from the effects of growth on ^{13}C incorporation. Finally, we hesitate to speculate about water-soluble storage. Known storage compounds tend to be insoluble, since this avoids osmotic imbalances when large amounts are accumulated. Trehalose is the only widely recognized exception, but its C storage function is debated.

2. Also was there a correction for extraction efficiency done for the CFE extracts?

CFE biomass is reported as “extractable biomass” throughout the manuscript, without correction. We consider this a more transparent approach, due to the large uncertainties in CFE extraction efficiency. We have added some theoretical background in the introduction, and a clarification of this point in the methods section:

“This method assumes a proportionality between extractable and non-extractable biomass” (line 61)

“CFE biomass is reported here as extractable biomass, without conversion with uncertain extraction efficiencies” (line 375)

3. To calculate C-growth with the ^{18}O method, increases in DNA are converted to units of C using the CFE derived MBC. How does DNA increase alone relate to the production of storage compounds?

This relationship is not evident in our data. A linear model fitted to DNA growth against labelled storage compounds has $R^2 = 0.15$ and $p = 0.12$. Furthermore, the conditions in our experiment (short incubation with resource addition) would tend to exaggerate this relationship – it would presumably be even weaker under conditions in which growth was under non-resource control, e.g. drought or hypoxic conditions, or maintained by storage degradation (e.g. starvation).

4. Also, could storage compounds be considered in the calculation to convert DNA production into units of C?

We do not think this a viable approach. The conversion of DNA growth to units of C relies on an assumption that the biomass of each cell (actually per genome) is constant. This is itself an approximation, but for storage there is really no reason to expect such a proportionality to hold in general. In fact, in the case of surplus storage one would expect an inverse proportionality, since

surplus storage occurs precisely when replication is limited by some other factor. This has been more clearly expressed in the revised manuscript:

“DNA-based measures of microbial abundance and replication also do not capture storage^{19,20}, since it is not expected to form a constant proportion of each cell’s biomass.” (line 67)

5. Line 58: replace “predicts” with “indicates”

This has been done.

6. Hypotheses: Hypothesis 3 is in my opinion directly related to hypothesis 1. I think they should be right after one another or could even be combined

Thank you for pointing this out. We have changed the order of hypotheses as requested. We agree that these are closely related, but standing biomass and growth under given conditions are conceptually distinct. We have emphasized this by adding “synthesis” to the hypothesis (line 154)

7. Line 82: delete: “the turnover of...captures”

Done

8. Line 86: change to: “supply and changes in element stoichiometry”

Done

9. Line 129: change to “ $24.7 \pm 2.5\%$ (mean \pm standard deviation) of the extractable microbial biomass”

Thank you for pointing this out. It is challenging to communicate this point without suggesting that the pool is a subset of the extractable biomass. We have reworded this and hope it is now clear:

“together representing a C pool of a scale equivalent to $24.7 \pm 2.5\%$ (mean \pm standard deviation) of the extractable microbial biomass C (MBC, by CFE; Figure 3)” (line 150)

10. Figure 2: Also add the letters that indicate the respective panel to (left) and (right)

This has been done

11. Line 305: You should mention here if you used a factor to account for CFE extraction efficiency.

Done, see Comment 2.

Reviewer 2

Major concerns include:

- the rationale of the study as important to ESMs,
- shortcomings of the figures,
- background on the extraction methods and storage compounds, and
- gaps in the methodology or explanation of the methodology

12. Rationale of the study as important to ESMs: While the authors discuss this finding as a major contribution to considering carbon use efficiency (CUE) and microbial biomass in Earth system process models, I think their more major impact is on considering how these storage compounds contribute to the stability, resistance, and resilience of microbial communities. My rationale for this is not that refining microbial CUE is unimportant, but rather that CUE in many process models is too coarse of a measure (it's not dynamic and/or emergent, but rather static and/or a fixed parameter) to be influenced by this new transitory pool. Because the authors find that these storage compounds are a transitional pool (almost like an intermediate in a chemical reaction), adding them into process models that operate at global scales might not have much impact...

What the authors should perhaps be suggesting is that microbial metabolism and the possible fates of metabolism are more accurately accounted for in models. I think the authors could suggest that these storage compounds be incorporated to process models that operate at a finer level of organization (e.g., community, or perhaps even ecosystem) and those that require a more accurate accounting of the resistance and resilience of communities and total biomass, but that this is unlikely to be accurately represented in models that operate at the Earth system scale.

We share this reviewer's doubts about explicit inclusion of storage into ESMs. It was not our intention to call for storage compound modelling at this level – our recommendation (original submission lines 223-224) was for a “reassessment of model inputs and interpretation of results wherever short-term measurements or dynamic changes are involved”. To avoid confusion on this point, we have revised the manuscript to reduce the emphasis on modelling at Earth-system scale and clarify the relationship to microbially-explicit modelling in general:

- On line 54 in the introduction, where we previously used “Earth system models”, this has been replaced by “models of the carbon cycle”. Our intention here was to highlight the broad importance of the biomass concept, rather than advocate storage modelling as such. We think “microbially-explicit models of the carbon cycle” accurately emphasizes the importance of microbial biomass concepts, while at the same time including the process-based models where Reviewer 2 agrees storage may be important.
- On line 258 we have replaced “required for C modelling and environmental management” with “required for understanding and managing ecosystem C balances”
- While storage might not need to be explicitly included as an ESM variable, the measurements of CUE that are used as inputs to large-scale models are often made on short time-scales that may well be distorted by storage compounds. We have added this explicitly in line 276:

“The important model parameter of carbon-use efficiency is typically measured over 24-hour periods³⁵, but over this time-frame we observed storage changes that constituted a substantial component of the microbial C balance.”
- Our results on storage compounds clearly demonstrate the shortcomings of the CUE concept that Reviewer 2 raises (coarse, static and not emergent), and this reviewer's suggestion of more nuanced consideration of microbial metabolism is an important point that we have included in the revised manuscript in line 278:

“This suggests that more nuanced representations of microbial metabolism and C allocation may be required to accurately account for microbial C use.”

13. Further, the authors suggest that these compounds should be considered part of the active biomass pool in ESMs. I also disagree on this point because in microbial-explicit models where the size of the biomass is consequential to process rates, biomass should be a proxy for the active number of cells. As such, whether there are many or few storage compounds should probably not be consequential to the active biomass pools nor should it affect process rates as reflected in ESMs.

We have removed reference to ESMs from the manuscript (see Comment 12), which we think largely resolves this concern.

On the question of definitions, please see response to Comment 56.

14. I'd suggest the authors consider a paper following on this more data-driven paper suggesting a refinement of CUE with these storage compounds, but also considering other non-biomass microbial C pools (like enzymes, antimicrobial compounds, secondary metabolites).

We appreciate this suggestion. In fact, we are currently considering how these experimental results can be integrated into a modelling framework to account for microbial storage, within the context of other extracellular C pools such as those mentioned, as well as extracellular polysaccharides for biofilm formation.

15. Moving forward with this paper without the reliance on claiming that the importance of this work is relevant to ESMs, the authors will need an alternate argument about the importance of their work. I'd suggest a larger reliance and deepening of the arguments about the importance of these storage compounds for the stability of microbial communities, like that currently discussed in lines 199-203. This was fascinating.

The revised manuscript avoids relating the results to ESMs (see Comment 12).

However, as Reviewer 2 points out, our results do have relevance to microbially-explicit models at smaller scales. We would therefore like to consider both aspects: implications for modelling (lines 273-279) and relevance for microbial stability (lines 303-312).

We appreciate Reviewer 2's enthusiasm for the second point, and have placed more emphasis on this in the revised manuscript. These points are set out in detail in our response to the following comment (16).

16. I'd like to see more separation of the concepts of the two storage strategies (i.e., surplus and reserve storage) and their impact on stability, resistance and resilience. Perhaps of service to this point would be a deeper contextualization in the discussion about what is known about the genetic regulation of TAG and PHB storage and if this matches the authors findings about the storage strategies associated with each, and with the authors questions about whether or not storage mechanism regulation differs among taxa or is relatively similar across all taxa in a community.

There has been considerable research into the biosynthetic pathways leading to TAG and PHB in various organisms. For TAGs these are multiple pathways, and the enzymes involved are not universally homologous. However, detailed work on genetic and metabolic regulation of storage biosynthesis has so far been limited to a few taxa of particular interest to bioprocess engineering

and medicine. There has also been no authoritative review that distils general conclusions about genetic control to the community level.

There is, however, some knowledge of conditions that promote storage synthesis, and we have integrated this into the manuscript to provide the context that Reviewer 2 has requested. In doing this, we have separated and emphasized the distinction between the two storage strategies and their relevance to resistance and resilience. We hope these changes address this comment appropriately:

- In Results and Discussion, surplus storage is discussed in connection with PHB, and reserve storage with TAG.

“These observations are consistent with PHB biosynthesis in pure culture²⁷, which is stimulated by excess C availability in diverse bacterial taxa²⁴” (line 174)

“A surplus storage strategy is particularly effective at buffering microbial activity by levelling out fluctuations in resource availability and stoichiometry¹¹. Furthermore, storage representing a substantial proportion of biomass offers a resource for regrowth following disturbance, indicating a potential role of storage in supporting resilience of this soil microbial community. In these ways, the resources stored in PHB could support the resistance and resilience of this soil microbial community against environmental disturbance⁴.” (line 181)

“One advantage of a reserve storage strategy is that strategic stores are assembled even under conditions of chronic resource shortage. This allows for bursts of activity to support, for example, reproduction or transition to a resilient starvation state⁴. Therefore, while reserve storage may be quantitatively smaller than surplus storage (reflected here in the lower amounts and changes in TAG relative to PHB; Fig. 2), it can help communities to persist under conditions of sustained stress, and even exhibit resilience against additional disturbances.” (line 222)

- We are hesitant to overstate the correspondence of surplus/reserve and PHB/TAG storage on the basis of this one experiment. Therefore we have also added a caveat sentence to stimulate some caution:

“There is no *a priori* reason to expect distinct storage strategies to correspond to different compounds, since both PHB and TAG can in principle provide C storage and mobilization under comparable conditions.” (line 245)

- We have deepened the discussion on stability:

“By enhancing the efficiency with which microbes incorporate transient resource pulses and supporting metabolic activity through periods of resource scarcity, storage can contribute to the survival of microbes facing stressful habitat changes. Resource availability in natural and agroecosystems changes over various time-scales, and we hypothesize that microbial storage may also be responsive to, for example, seasonal changes in belowground C inputs, supporting microbial activity through resource-poor winter periods or dry summers. Moreover, storage enables a diversification of resource-use

strategies, reflected here in the contrasting responses of PHB and TAG. Ecosystem stability is promoted by diverse strategies within the community³⁸, suggesting that storage can contribute to resistance and resilience of microbial communities facing environmental disturbances.” (line 303)

- The concluding sentence, which previously pointed more toward modelling, now emphasises ecosystem responses to change:

“can enrich our understanding of microbial resource use and its contributions to biogeochemical cycles and ecosystem responses under global change” (line 333)

17. When talking about nutrient limitations, showing C:N ratios rather than (or in addition to) DOC and TDN would be useful. For example, the point the authors are making in line 102 would be aided by a comparison of C:N in biomass and SOM, or at least a figure of soil nutrients.

Thank you for this suggestion. Dissolved C:N ratios (DOC/DN) have now been included in Fig. S1 to aid this comparison, and microbial biomass as a new Fig. 3. This has also been integrated into the main text discussion:

“This early decline in mineralization was consistent with the onset of nutrient limitation, after microbial growth on the added glucose had depleted easily available soil nitrogen and driven up the C:N ratio of dissolved resources (Supplementary Figure S1)” (line 119)

We would be happy to move some of these supplementary figures into the main text if this is preferred. However, we tried to keep Section 2.1 as concise as possible to move quickly on to the storage results (Reviewer 3: Comment 82)

See also Comment 24

18. Also, figure S3 does not support the statement in line 113 and in fact seems to be contradictory.

The original phrasing was unclear. This has been revised:

“With high-C addition after 24 h, DOC was far lower with nutrient supplementation than without (Cohen’s $d \gg 1$, family-wise $p < 0.001$), and DOC level for this treatment did not change further to 96 h, despite having higher N availability at 24 h than the no-nutrient treatment (Cohen’s $d \gg 1$, family-wise $p < 0.001$). This indicates that the microbial community had depleted the added C and re-entered C-limited conditions.” (line 134)

19. The error on the green and yellow figures is also unclear. Is the error for both bars? Or is there other shading I can’t see? Error bars (as used on other stacked bars) are easier to interpret.

From the reviewer comments it was clear that the shading approach was not effective in communicating the uncertainty. We have added error bars to the figures as requested.

20. Lastly, I'd really like to see the posthoc differences on the figures. It's hard to know where the statistical differences are by looking at the figures.

We have added posthoc differences as requested, and updated the methods section to specify the underlying statistics.

21. the introduction lacks a bit of context and should be refined to include (1) information about what different extractions methods do and don't extract/show ...

We have expanded the introduction of the biomass methods to ensure that this is clear, also for the general audience:

"Conventional measurement of soil microbial biomass uses fumigation with chloroform to lyse cells, followed by extraction of the released biomass into an aqueous solution for measurement (chloroform fumigation-extraction, CFE)¹⁴. This method assumes a proportionality between extractable and non-extractable biomass¹⁵. Other measures in widespread use are proxies such as cell membrane lipids or substrate-induced respiration¹⁶⁻¹⁸. Only CFE provides biomass in units of C, however, and these other methods are typically calibrated against it." (line 58)

"DNA-based measures of microbial abundance and replication also do not capture storage^{19,20}, since it is not expected to form a constant proportion of each cell's biomass." (line 67)

22. (2) ...information about the storage compounds

We have added additional information about the storage compounds to the introduction:

"These are both hydrophobic lipids that are stored as inclusion bodies in the cytosol (i.e., intracellular lipid droplets)⁵. PHB is a high-molecular-weight polyester of β -hydroxybutyrate, while TAGs consist of three fatty acids (of diverse structures) esterified to a glycerol backbone⁴." (line 44)

23. text explaining what is observed with conventional biomass extraction methods vs. these TAG and PHB methods would help to support lines like 153 (that this is the first study to show microbial storage dynamics in a terrestrial ecosystem).

See response to Comment 21

24. the observations of shifting C:N in biomass relative to soil/solution C:N seem consistent with previous observations of biomass responses to changing nutrient conditions (i.e., C storage during nutrient scarcity (increase in C:N), turnover during nutrient abundance). However, the authors claim that the previous measures wouldn't include these storage compounds. I was left wishing for more detail on what exactly is extracted by each type of extraction so I could understand whether this was an appropriate parallel or not.

We have provided more background on the different extractions (see Comment 21), highlighting that chloroform fumigation extraction relies on aqueous extraction, while the storage compounds measured here are water-insoluble lipids.

The distinction between the microbial storage compounds and the extractable microbial biomass measured in the CFE method is most clearly evident from the different dynamic behaviours of these pools, which the revised manuscript now highlights better:

“high C input stimulated a large increase in PHB, particularly when not supplemented with nutrients (a 308% increase over the zero-glucose, no-nutrient treatment at 96 h, with Hodges-Lehmann median difference of 36.0 – 42.9 $\mu\text{g C g}^{-1}$). In comparison, extractable biomass reflected a non-significant mean difference of only 33% between these treatments” (line 161)

“assimilation of glucose C into new PHB continued between 24 and 96 h under the nutrient-limited conditions of the high-C, no-nutrient treatment (Hodges-Lehmann median difference of 10.2 – 13.3 $\mu\text{g C g}^{-1}$, 95% confidence interval), while extractable microbial biomass C showed no significant change.” (line 167)

The first author has recently published a comprehensive review of storage in soil, and is not aware of previous studies demonstrating that shifts in extractable microbial biomass C:N ratios are caused by accumulation of storage compounds. If key literature has been overlooked, we would be very keen to take this into account. We are convinced that storage forms remain to be discovered, and that C storage is a widespread response to increased C availability in soil, but we have some doubts about whether soluble C will turn out to be a major storage pool, given the osmotic implications (see also response to Comment 1).

A new figure of microbial biomass C:N ratios is now provided as supplementary Fig. S3 to enable the interested reader easily make these stoichiometric comparisons for themselves.

25. (3) it's important that the authors tell us more about the structure and solubility of these compounds, as well as where (in, on, or around the cell) these compounds are stored in order for the reader to fully get the whole picture and understand how we can structure this new information within the scaffold of what we know (like extraction of biomass into polar solvents like water and K₂SO₄).

This information is provided in the revised manuscript (see Comment 22)

26. PHB and TAG should be explained as individual compound (classes?) in order to provide rationale for keeping their results separate in figure 2.

We hope that this has been addressed under Comment 22.

27. Gaps in the methodology or explanation of the methodology: 1) I am concerned about the comparability of the two incubation experiments given the large difference in biomass between the 18O incubations and the primary incubations. Ecological theory (species area relationships) predicts that the reduction in incubation size may lead to greater stochasticity in the diversity and community composition of the smaller microcosms and it's unclear how these differences are likely to impact the biomass and growth rate results they observe, therefore, the authors should provide evidence that the reduction in size of the 18O microcosms did not substantially influence their results between the two incubations

We appreciate this concern and gave it consideration during the experimental design. Unfortunately the high cost of ¹⁸O-labelled water, combined with the large amounts of soil required for sensitive measurement of storage compounds, makes it infeasible to measure growth and storage compounds in the same microcosms. This mis-match of methodological scales also makes it impossible to prove conclusively that these measurements really are unaffected by the different scales. However, we are confident that this is the case, for reasons now set out in the revised manuscript:

“This smaller scale was necessitated by the cost of ^{18}O -water. This is nevertheless larger than the soil amounts typically used for DNA extraction, which achieve consistent measures of bacterial and fungal community composition. This is also orders of magnitude larger than the scale of microbial interactions⁴⁴. These considerations, alongside the care taken to ensure identical conditions of temperature, moisture and handling, give confidence that this incubation was representative of the same processes occurring in the larger microcosms.” (line 416)

Unpublished tests in our labs also show that small soil samples yield reproducible CO_2 efflux rates, indicating that 0.5 g of sieved soil still effectively averages over micrometer-scale variation in microbial activity.

28. 2) I found it difficult to understand exactly how the authors calculated various percentages and ranges of carbon contributions. I think this could easily be clarified through more detailed methods (potentially supplemental if space is limiting).

We were unsure of exactly which aspects of the calculations were unclear, but we have endeavoured to improve any points of potential confusion, as follows:

- In the methods section:

“Results for CO_2 , MBC, DOC, DN, TAG, PHB, and isotopic compositions were calculated for each independent sample and reported as mean \pm standard deviation for each treatment group, unless otherwise noted. Comparisons between these pools were similarly calculated at the sample level before expressing as mean \pm standard deviation.” (line 441)

“Ranges for treatment effects on DN, DOC and MBC reported in the text reflect 95% family-wise confidence intervals from pair-wise Tukey’s HSD tests.” (line 448)

“Analysis of storage compounds (PHB and TAG) proceeded by robust ANOVA of medians for each timepoint separately using the R package WRS2⁴⁸. Consistent with the median-based robust ANOVA, storage differences between treatments reported in the text are median differences, with uncertainty given as 95% confidence intervals calculated by the Hodges-Lehmann estimator (R package DescTools⁴⁹)” (line 445)

“The corresponding mean extractable microbial biomass values were applied to convert to absolute units of $\mu\text{g C}$, using standard rules of error propagation⁵¹, to provide the DNA-based measure of mean microbial biomass growth for each treatment. These DNA-based growth estimates were combined with the mean production of labelled storage compounds (sum of C in glucose-derived PHB and TAG), again using rules of error propagation, to obtain estimates of total (DNA-based and storage) mean biomass growth and associated standard deviations. These were subjected to 2-way ANOVA and Tukey HSD to test the significance and size of treatment effects (Fig. 3). Arithmetic comparisons between MBC, growth and storage pools (for example, the relative scales of DNA-based growth and storage growth) were calculated using mean values with error propagation.” (line 464)

- With respect to the share of biomass, we have reworded this (Comment 9).

29. Ideally the authors could provide code supporting their calculations, but at the very least they should provide the equations used in these calculations

We have provided our code and source data as supplementary material.

All source data and code will be published on the Zenodo open data repository under DOI 10.5281/zenodo.6386047 immediately after publication. This is now reflected in the data and code availability sections.

30. 1 – suggest that the title be changed to “Non-replicative carbon storage by microorganisms is an overlooked pathway of biomass growth”

We would prefer not to change the title in this way, because this would imply the existence of “replicative carbon storage”. In the first author’s recent review (Mason-Jones et al., 2022), we propose to define storage as “accumulation of chemical resources in a particular form or compartment, in order to secure their availability for future use by the storing organism”. Underlying this concept is that storage is not formed for an immediate purpose, but rather for its future value to the same organism. “Replicative storage”, in the sense of carbon invested in replication, would (a) provide an immediate rather than future fitness benefit, and (b) would not represent usage by the same organism (although we acknowledge that there can be a grey area here with microbes). In other words, it would not be storage by our definition.

We have included the full definition in the revised manuscript to make this point clearer.

31. 25-26 – these concepts really come out of nowhere. While I really liked these in the manuscript and am suggesting even heavier reliance on them overall, they need more build up in the abstract

We now introduce environmental change earlier in the abstract and come back to it in the context of resistance and resilience. We hope that this has adequately addressed this concern.

“Resource investment in storage allows microbes to decouple their metabolic activity from immediate resource supply, supporting more diverse microbial responses to environmental changes” (line 18)

“...and an underlying mechanism for resistance and resilience of microbial communities facing environmental change” (line 27)

32. 38 – As mentioned in the overview, more information on the structure and location of these compounds is needed. While there are likely many places in the intro that this can go, I like after line 38 could be good.

See Comment 22

33. 42-51 – This paragraph would be a great place to include more context about what compounds are extracted with the biomass methods. Line 45-46 about the DNA-based methods should be more direct.

See Comment 21.

We have also revised the sentence on DNA-based methods:

“DNA-based measures of microbial abundance and replication also do not capture storage^{19,20}, since it is not expected to form a constant proportion of each cell’s biomass” (line 67)

34. 48-51 – consider making this the topic sentence

Thank you for this suggestion. We have inverted the structure of this paragraph to achieve this revision (line 53).

35. 53-60 – More details on reserve storage would be helpful, as it’s less intuitive than surplus storage. For example, what do the culture studies find? Do we know this from one microbe, or are there culture studies across a wide range of bacteria? Also, it’s unclear from the hypotheses whether this study can and will be able to unravel these. Are the experiments truly set up to test this?

Please see Comments 16 and 53.

36. Line 63-64 – This hypothesis is wishy-washy and weak. What is meant by “a substantial portion?” How can this be tested?

This has been revised to:

“Microbial storage compounds are a quantitatively important pool of soil microbial biomass under C-replete, nutrient-limited conditions.” (line 81)

In our view it is reasonable to express the hypothesis without rigidly defining “quantitatively important”, even though this creates a potential grey area. Most readers would agree that our data supports this hypothesis, even though we did not define a threshold for testing the hypothesis. However, if strongly preferred we could add a post-hoc threshold like “>5%”, this being the range at which we would no longer have considered it of quantitative importance.

37. Line 66 – Replace “complementary” with more specific language

Done

“nutrient supplementation (N, P, K and S) will suppress” (line 86)

38. Line 68 – I challenge the concept that these are biomass, particularly in the context of this study showing that these compounds are a transient pool.

Please see Comments 56 and 59.

39. 73 – this phrasing implied to me that these nutrients were tested separately rather than in tandem. Please amend.

Thank you for pointing this out.

“A combined nutrient treatment (N, P, K and S) provided inorganic fertilizers common in agriculture” (line 92)

40. 76 – predicted by what? C:N? Please include assumptions.

This has been clarified:

“based on microbial biomass C:N:P ratios typical of agricultural soil²⁶ and an assumed C-use efficiency of 50%” (line 96)

41. 77-79 – where these measurements isotopically-enabled? I presume yes, but please include.

This has been clarified:

“CO₂ efflux and its isotopic composition was monitored at regular intervals” (line 97)

42. Results and discussion – please flesh out surplus vs reserve storage concepts and how this work addresses these mechanisms.

See Comment 16

43. 95 – it is unclear what “these” is referring to

We have revised as follows:

“Nutrient supplementation barely affected CO₂ efflux rates from the zero- or low-C additions and for none of the zero- or low-C treatments was N availability (measured as DN) significantly reduced relative to the control at 24 h” (line 113)

44. 105 – I don’t see a decline in TDN between high C no nutrient conditions between 24 and 96 h in this figure. This should be refined or clarified. Does this conflict with line 112? I am finding this confusing to follow.

The decline was relatively small and not essential to the discussion. Therefore we have removed this statement to avoid any confusion.

45. 113 – I recommend being very specific about which control is being referred to here, as it is confusing to interpret the figures. It appears this comparison refers only to the no-nutrient addition incubation, but that doesn’t align with the graph. When I look at Figure S3, I see that the DOC without the N+P is greater than with N+P. But this line says the opposite (I think, but the wording is confusing). This needs to be fact checked and/or re-worded. More generally, I think the authors should include the calculations when appropriate in the methods (or an extended supplemental methods) section. This would greatly clarify for the reader exactly what is being compared to what.

This has been more revised to clarify:

“For this high-C, nutrient supplemented treatment, dissolved N decreased only moderately over 24 h (56.2 – 97.9% relative to the zero-C, no-nutrient treatment)” (line 132)

“With high-C addition after 24 h, DOC was far lower with nutrient supplementation than without” (line 134)

We have gone through the methods to clarify any comparisons that might cause confusion. Calculation code and source data is provided in the supplementary materials.

46. 130 – what is the ratio being referred to? I think you have added together the PHB and TAGs and then taken the ratio of that value and the MCB ug C/g here, but it is not entirely clear from the text. I recommend specifying these and other calculations in either the methods, or an extended supplemental methods section.

We have reworded these sentences to clarify which percentage is meant. We hope this has resolved the confusion:

“PHB and TAGs were both found in the control soil (zero-C, no nutrients after 24 h; **Error! Reference source not found.**, A&C), together representing a C pool 0.25 ± 0.03 (mean \pm standard deviation) times as large as the extractable microbial biomass C (MBC, by CFE; Figure 3). This ratio of stored C (PHB + TAG) to extractable MBC ranged from 0.19 ± 0.02 to 0.46 ± 0.08 over all treatments” (line 149)

See also Comment 9.

47. 139 – include a p-value

This has been added.

48. 142 – to be consistent, use the term zero-C rather than control

This has been revised accordingly.

49. 154-156 – it is unclear where the data is to support this assumption. Please provide a reference or details of how this basal respiration period was calculated.

This has been clarified:

“At the end of the incubation, stored C across the various treatments was sufficient to support 109 – 347 h of microbial respiration at the CO₂ efflux rate of the zero-C, no-nutrient treatment (i.e., basal respiration). Much longer periods would be envisaged if accompanied by strong downregulation of energy use in response to the stress²⁸.” (line 177)

50. 180 – For the PHB:TAG comparison please specify units in text (is it in % of total ug C/g or a simple ratio of ug C/g of each storage compound?).

This has been provided:

“Notably, the relative allocation of glucose C between PHB and TAG remained relatively constant (PHB:TAG ratio of glucose-derived C ranged between 7.0 and 11.5 across all treatments)” (line 217)

51. Paragraph beginning at line 195 – I would find it helpful and more appealing to a wider audience, if there was a discussion about what is known about the genetic regulation of PHB and TAG genes in pure culture or other systems, and if these patterns of regulation are consistent with the data presented here. For example, does the genetic regulation of PHB or TAG suggest that they are used for different storage purposes? Does regulation of these two gene cascades seem to be similar or vary a lot among organisms? (Note: I’m not suggesting the authors carry out a separate genomic analysis, but mining the literature for this information would lead to a richer discussion of the implications of these findings).

See Comment 16.

52. 198 – clarify utilize... synthesize? Degrade? typo in the spelling of “fulfill”

This has been corrected to “synthesize” and the correct spelling. (line 248)

53. 199-203 – this is fascinating. I’d love this to be more central.

We have considerably expanded the coverage of reserve versus surplus storage in the introduction and discussion – see also Comment 16.

54. Section 2.3 – Given that the 18O experiments were performed on significantly less soil than the storage incubation, I’m concerned that the two incubations are not comparable. Even for well-homogenized samples, by random chance and from ecological theory (species area relationships), we expect that smaller incubation should have large differences in diversity and turnover between replicates simply due to ecological drift, therefore, I think it is important for the authors provided evidence that the small incubation are a good proxy biomass and growth rate for the larger incubations despite these differences in community structure.

See Comment 27

55. 211 – add “DNA-based” in front of 18O to add clarity

This revision has been made

56. 207 & 222– I fail to be convinced that these storage compounds should be considered biomass rather than microbial products because in models and in reality, biomass pool size regulates C turnover due to its action on other C compounds. These storage compounds have no agency, so until they are incorporated into structures that can act on SOM, I argue they should not be considered biomass.

A measure of microbial biomass must of course be suited to purpose. In our opinion biomass should be seen as a measure of material stock, e.g., mass of C. This is now more clearly stated in line 36-37. In this case, an intracellular, metabolically available cellular component should be included, as this clearly constitutes a proportion of the C pool contained within the organism. Whether or not total community biomass is a good proxy of microbial abundance or activity is a crucial question (and our results argue against this). However, in our view the term should not be redefined in order to make it a good proxy. This would be undesirable because the biomass concept is used in various other related settings in which an accurate “mass” definition is important, for example ecological stoichiometry, or carbon balance calculations.

57. 218 – rather than C limitation, do the authors mean in the low C addition? Or do you know where a threshold of C limitation is? Even with no C added, there isn’t necessarily a C limitation. I’d argue that C limitation (here or in the intro) need to be defined and that the level at it occurs in this specific soil should be discussed.

Defining an absolute threshold of C limitation is challenging. In the manuscript section 2.1 we present various lines of evidence to demonstrate which limitations prevailed at particular times and in response to particular treatments, and this is important for the fullest interpretation of our results. To make this clearer we have changed the title of this section and added a first line to make this link explicit:

“We first describe observed patterns of soil respiration and dissolved nutrients that aid interpretation of the prevailing resource limitations during storage compound synthesis and degradation” (line 110)

We have also specified the corresponding treatments at this point in the manuscript to make this clear:

“Even under conditions of C limitation (zero and low-C treatments)” (line 269).

58. Line 222 – I appreciate the authors caveat here that this is likely to be most significant for short-term dynamics.

Thank you – yes we think this is an important point which has been retained in the revised manuscript.

59. 236 (and as discussed elsewhere) – Given that this data shows this is transient, I take issue with this being called a growth pathway. Instead, perhaps it should be considered a carbon allocation strategy.

It is not clear where our data proves the transience of storage compounds, since large amounts remained in the biomass at the end of our experiment. However, from theory we agree with the Reviewer that they are transient pools. We don't see permanence as a conceptual requirement for biomass or biomass growth, however: from cell membrane lipids to leaves on trees, many (arguably all) parts of biomass are transient over some finite timespan.

Reviewer 2 is entirely correct that it is a carbon allocation pathway: it is carbon allocation to storage biomass (in our view, storage compounds should be considered a component of biomass; see Comment 56). Since growth in the context of our manuscript refers to the formation of new biomass (clarified in line 36-37), we disagree on this point and think that formation of storage compounds is indeed a type of growth.

60. Figure 1 – a complementary figure showing the % CO₂ that was labeled (rather than the current inset) would help so the reader doesn't have to eyeball this.

We have made this change to Fig. 1 and are grateful for this excellent suggestion.

61. Also, please show the x axis (for both the main and the labeled) in 12 hour increments to match the text and align with the harvests. Also, the y-axis in the main panel should be labeled "Total CO₂ rate" to distinguish it from the "Labelled CO₂ rate" inset. It was unclear to me originally that the main figure represented both labelled and unlabelled fractions.

We have made the requested changes, although we found that 12-hours increments made quite a cluttered diagram and therefore opted for 24-hour increments instead. We hope this is acceptable. Please note that the figure does not include a point at the 96-hour x-axis value. This is because the CO₂ data represents cumulative efflux to each sampling point, and it is conventional to present the mean rate values at the midpoint of the sampling interval. This is explained in the caption accompanying Fig. 1.

62. **Figure 2** – the x axis labels should be less redundant so people only looking at figures can understand them. For example, both the glucose and nutrient treatments include a "none" to explain them. Also, since the nutrient addition wasn't just N+P, maybe this should be coded differently.

The axis labels have been improved accordingly.

63. Also, the error (shown as shading) is unclear. Is the shading just for the green or also the yellow? I'd suggest doing the error bars as in figure 3 or separating the components into separate figures.

From the reviewer comments, it was clear that the shading approach was not effective in communicating the uncertainty. We have added error bars to this figure as requested.

64. Additionally, this figure might be improved by adding an indicator (perhaps a star or dashed horizontal line) of TAG and PHB levels to the chart. This would help clarify the discussion of the ratios of stored carbon to biomass carbon on line 128.

We hesitate to include this, as the revised figures already carry a lot of detailed information, including error bars and pairwise comparison indicators. This would also be challenging because of the different scales of the y-axes between the two storage compounds. Furthermore, readers can make a qualitative comparison of the two storage compounds from the axis scales and in Fig. 4. However, if the reviewer feels strongly about this, we would be prepared to revise accordingly.

65. I'd also suggest showing the post-hoc differences on the figure to make this more informative.

This is included in the revised figures.

66. Figure 3 – I suggest adding a top panel here with the chloroform fumigation data (from S2) because I needed to do some eyeballing to support line 176.

The MBC results are provided in the revised main text as a new Figure 3.

67. Supp Figure 1 – state that there is no statistical difference or show that. Also, showing the C:N ratio would help support the point in lines 97-98. Also, the caption needs more details (and punctuation). It's not clear from the caption what the figure describes.

Statistical comparisons are now reflected in this figure, and we have added the corresponding C:N ratios as suggested. The caption has been wholly revised to provide more information.

68. Supp Figure 2 – change carbon to glucose

This figure has been revised accordingly (now Fig. 3 in main text).

69. Supp Figure 4-6 – These figures are inconsistent with the formatting on the other figures. I would appreciate it if these were brought into alignment, but at a minimum the authors should define what the colors represent in the captions. I assume Glc refers to glucose, and that GZ, GL, and GH are representing the different glucose treatments. But this was not easily or immediately clear. In addition, it's unclear which compounds are important on these figures. It would also be helpful, if the fungal and bacterial biomarkers referred to in the text were indicated on the x-axis of the graphs in S4 and S5, as the x-axis names differ slightly from those used in the text. Perhaps detail can be added to the caption.

We apologise for using unfamiliar treatment codes in these figures and have amended this. Furthermore, we have combined S4 and S5 into a single new Figure S4. The new figures S4 and S5 adhere more to the formatting of the other figures. Precisely following the data presentation of the main text figures (e.g. Fig. 2) is difficult with the data in Fig S4, since this would require separate plots for each fatty acid, whereas the intention is to provide an overview of the fatty acid profiles for different treatments (i.e. to show all fatty acids on one plot). The biomarkers referred to in the text are now highlighted in different colours on the axis, and this is explained in the caption. The nomenclature of the fatty acids in the text and supplementary materials has been brought into alignment.

70. Supp Figure 6 – Are these glucose-derived? Why a new figure format?

This has been clarified in the caption, and the figure fully revised to the consistent format (new Fig. S5).

71. Line 274 – How long was the transect?

This has been added:

“Five samples along a 50 m field transect” (line 339)

72. Line 276 – How long was the soil stored at 4 degrees?

This has been added:

“Soil was stored at 4°C for one week prior to sieving” (line 341)

73. 283 – please include nutrient concentrations

This has been added:

“(combined $(\text{NH}_4)_2\text{SO}_4$ and KH_2PO_4 , respectively 0.613 and $0.106 \mu\text{mol g}^{-1}$ soil)” (line 347)

74. 284 – it is unclear whether ^{13}C and ^{14}C were added together and why

This has been clarified:

“The ^{14}C label in the added glucose enabled rapid and accurate measurement of glucose-derived C in liquid extracts by scintillation counting” (line 352)

75. 286 – I’m a little confused about how this ratio was determined for the 0 carbon added treatment. More details about the stoichiometric ratios of C:N:P in each treatment would be helpful. Was the same concentration of nutrients added in the low C treatment as in the high carbon treatment? Please clarify in text.

The text has been updated to make this clearer:

“The same amount of nutrients was used in all nutrient-addition treatments, with this set to be sufficient for the complete utilisation of all C added in the high glucose treatment” (line 354)

This was a point of considerable discussion during the experimental design. The chosen approach has the advantage that the nutrient addition is a constant and independent factor, in other words the same total change in nutrient availability. It has a drawback that the stoichiometry of additions differs between the treatments. We decided against a constant-stoichiometry approach because the only the stoichiometry of additions can be controlled, whereas it is actually the stoichiometry of all available C and nutrient sources that constrains microbial activity, which is difficult to accurately determine. Therefore, a constant-stoichiometry experiment would be harder to interpret because neither the nutrient addition nor the true stoichiometry of available resources would have been well defined.

76. 294 – I’m unclear about the reasoning behind the 24h vs 96h time choices, it would be nice if the authors included some insight into this choice.

This is clarified in the revised manuscript:

“Microcosms were harvested after 24 and 96 hours, with these incubation times selected to balance the synthesis of storage (previously observed over a timeframe of days⁸) with the risk of artefacts induced by recycling of labelled biomass¹⁹” (line 98)

77. Line 310 and Line 326 – Please add to the text the number of grams of soil used for the PHB and TAG analyses.

This has been added:

“extraction of 4 g freeze-dried soil into chloroform” (line 381)

“extracted from 5 g frozen soil” (line 398)

78. 327 – 328. End sentence with “in soil.38” and start new sentence with “Lipids were first...”

Done

79. Line 345 – I haven’t done this kind of incubation or this small of an incubation, but I’m not sure that an incubation of 0.5g of soil in 2mL tubes is comparable to the 100mL microcosms containing 25g of soils. Would there be enough room for fungal proliferation/spatial distributions that can occur in the larger tubes to occur in the smaller? Additionally, would they not dry out during the pre-incubation? Were they also subject to the pre-incubation? Generally I’d just like more details here and some proof that the 2ml Eppendorfs equivalent to the larger microcosms

See Comment 27

80. 359 – please show mixing models and a definition of end members

This information has been provided in detail in Supplementary B, with a reference to this at the recommended position in the text (line 434). This is a standard calculation for isotopic labelling studies and not of particular interest to a broader readership, so we felt that this fitted better into the supplementary materials. However, if the reviewer has a strong preference for its inclusion in the methods of the main text, we would be prepared to place it there instead.

Reviewer 3

81. **Key issue 1:** the authors do not very clearly spell out the broader implications of accounting for storage compounds in soil microbial growth/biomass. While this may be very clear to soil microbial ecologists/soil biogeochemists, it may not be to a broader readership. This could be expanded on both in the Introduction and much more thoroughly in the Discussion.

Thank you for encouraging these improvements. The revised manuscript addresses the relevance and broader implications in a number of locations.

In the abstract we now provide a clearer statement on the relevance of microbial storage in general:

“Resource investment in storage allows microbes to decouple their metabolic activity from immediate resource supply, supporting more diverse microbial responses to environmental changes” (line 18)

In the introduction we have made the biogeochemical relevance more explicit:

“Microbial storage could substantially influence microbial fluxes of C and other nutrients¹¹, changing our understanding of soil biogeochemical fluxes and their response to environmental changes.” (line 50)

We now make an explicit call for advancement of methodology on the basis of our results:

“calls for methodological advancements to more systematically capture these (and possibly other) storage compounds in assessments of microbial growth.” (line 157)

With respect to ecological relevance, In the discussion we have expanded on the implications of storage strategies on resistance and resilience of soil microbial communities (see Comment 16).

This comment is closely related to Comment 86 – please see also the revisions made there.

82. **Key issue 2:** the entire first section of the main text (Section 2.1) does not feel very relevant to the point of the paper, which is about the role of storage compounds. If the authors were to keep that section, I would suggest moving into supplementary, as well as Figure 1. As it stands, it is confusing why the authors jump into a discussion on CO₂ efflux under different nutrient conditions on a paper whose main take home is supposed to be about storage compounds.

We would like to keep this section, as these more commonly measured aspects of microbial activity are crucial for interpreting the subsequent storage measurements. However, we appreciate that this might not be immediately clear for a reader. Therefore, we have added a new opening sentence to make this explicit:

“We first describe observed patterns of soil respiration and dissolved nutrients that aid interpretation of the prevailing resource limitations during storage compound synthesis and degradation.” (line 110)

83. **Key issue 3:** The paper felt a bit thin on data, even if some interesting results are presented about the role of TAG and PHG. At core, its really two main data figures (Fig 2 and 3), especially if Figure 1 was moved to Supplementary.

We have tried to present our findings as concisely as possible, showing the key results necessary to convincingly support our conclusions, as Reviewer 3 has noted in their preamble. We can state confidently that our concise presentation does not reflect a lack of data: The incubation involved over 100 microcosms, with over 1000 data points across ten measurement variables (MBC, PHB, TAG, ¹³C, ¹⁴C, CO₂, DOC, DN, DNA, ¹⁸O).

84. **Key issue 4:** it is unclear why the incubation was relatively short and why the particular storage compounds TAG and PHG were exclusively measured. There may be good reasons for both, it was just not clearly spelled out to me.

The selection of the incubation times is now explained at the end of the introduction:

“Microcosms were harvested after 24 and 96 hours, with these incubation times selected to balance the synthesis of storage (previously observed over a timeframe of days⁸) with the risk of artefacts induced by recycling of labelled biomass¹⁹.” (line 98)

The focus on TAGs and PHBs is now justified in the introduction:

“These two C-rich storage compounds are of particular interest as they are accumulated by diverse microbial taxa⁴ and methods are available for their measurement in soil^{7,8}.”

85. **Key issue 5:** I found the final figure to be confusing. I think this presents a good opportunity to clearly spell out terms used in the paper, differentiate different hypotheses, and show results, and I think I could do that more effectively.

Thank you for pointing out the difficulties with this conceptual figure. Please see Comment 113.

86. Overall, I think you need to make a stronger case here for the broader applicability of the findings in your paper. It doesn't seem the main selling point should be that it helps explain the "mechanisms underlying resistance and resilience of microbial communities" (i.e., last line of the Abstract) as the paper doesn't really focus on this, but rather should focus on how it is essential to our accurate understanding of soil carbon cycling.

We agree with Reviewer 2 that the implications for resistance and resilience are of great relevance. We have expanded more on this aspect to justify it as a key implication of the paper (Comments 12 and 16).

However, we also agree that the implications for our understanding of soil C cycling are important, and have emphasized this more in the revised manuscript.

We now note the relevance for a crucial parameter of microbially explicit soil C models:

"The important model parameter of carbon-use efficiency is typically measured over 24-hour periods³⁵, but over this time-frame we observed storage changes that constituted a substantial component of the microbial C balance. This suggests that more nuanced representations of microbial metabolism and C allocation may be required to accurately account for microbial C use." (line 276)

We note how storage questions the appropriateness of the common assumption of "overflow respiration" in soil C models:

"Models of microbial growth typically assume that increases in biomass match the elemental stoichiometry of the total biomass (the assumption of stoichiometric homeostasis³⁶), and therefore implement overflow respiration of excess C under conditions of C surplus³⁷. However, substantial incorporation of C into otherwise nutrient-free PHB and TAG clearly does not follow whole-organism stoichiometry." (line 293)

We note how C storage also relates to retention of C as well as other nutrients in the soil:

"The short experimental timeframe here is representative of environmental resource pulse and depletion processes, such as the arrival of a root tip in a particular soil volume or death and decay of a nearby organism. Storage provides stoichiometric buffering during such transient resource pulses, which is predicted to increase C and N retention over the longer term¹¹." (line 299)

87. 21: What do you mean they accounted for 20-46% of extractable biomass? I thought these storage compounds were not extracted in the chloroform fumigation method?

We realised that this wording was confusing and have revised this to be clearer:

"Together these compounds comprised a C pool between 0.19 ± 0.03 to 0.46 ± 0.08 times as large as extractable soil microbial biomass" (line 23)

We have also added more information on what different biomass measurements represent (see Comment 21).

88. And in line 130, you state that storage C is 20-46% as large as the extractable biomass pool? Maybe you mean the same thing here, but it is confusing to me. I would state the size of this pool of storage C relative to other estimates.

We have reworded this, and hope it is now clearer:

“together representing a C pool 0.25 ± 0.03 (mean \pm standard deviation) times as large as the extractable microbial biomass C (MBC, by CFE; Figure 3)” (line 151)

89. 30: This opening sentence feels like its lacking something – specifically, why is this important? Isn't it important because it determines the flow of C and other nutrients through these systems?

We have revised this to:

“Microbial assimilation of organic resources is crucial to the flows of C and other nutrients through ecosystems.” (line 33)

90. 37: Why highlight PHB and TAG? Are these the most quantitatively significant? The ones we are best at isolating? The most widespread? It feels a little early in the paper to give such specifics, especially with no context for why you are targeting these two specific compounds. I would perhaps save this for a later paragraph, and give me context as what these storage compounds are, and why you are focusing on them.

The focus on these storage compounds is now explained in the text (see Comment 84).

We have also added additional information on the nature of these storage compounds (see Comment 22).

91. 41: I still think this needs a stronger sell for why this is a major oversight that needs correcting. What is inaccurate in our current estimates of microbial processes and C cycling that would change by accounting for storage compounds?

See revisions made for Comment 81

92. 49: What do you mean by ‘local ecological stoichiometry’? I find this confusing. Can you more clearly state here examples of where biomass growth is important to know about? I definitely think microbial-explicit models is one good example.

We have rephrased this to be clearer, and also added two other perspectives from which growth is an important biological process.

““Biomass growth” is a cornerstone concept at scales from the ecological stoichiometry of individual cells to microbially-explicit models of the C cycle^{12,13}, and for defining the nutrient demands of organisms and their productivity¹².” (line 53)

93. 51: Any good reference to cite here?

A reference has been added (“understanding of C assimilation and utilisation^{4”}) (line 58)

94. 52: I would add “interpretation of microbial storage patterns”

Done

95. 53: I would italicize 'reserve storage' and 'surplus storage' or put in quotes, the first time you use them. It could also be very useful to show a simple conceptual figure here of definitions and predictions, as your Figure 1.

These terms have been previously introduced and discussed in detail in Mason-Jones et al. (2021) and Manzoni et al. (2021). The core message of the current manuscript is the quantitative significance of storage compounds and their significance as overlooked forms of microbial growth. Storage strategies, including the concepts of surplus and reserve storage, are very useful to interpret storage patterns and understand their biological drivers. However, we would prefer not to lead with these at the forefront of our message, since they really play a more interpretative role in this work.

96. 66: "At the community-scale ..." Not sure what this means. Compared to what? Individual? Population? Or compared to reserve storage? Not sure what you are comparing here.

This has been revised for clarity:

"surplus storage is likely to be quantitatively more important than reserve storage when measured across an entire soil community" (line 85)

The point we want to make is that this might not be true at the species level, where specific strategies might lead to high levels of reserve storage. But we would make this prediction for measurements across whole soil.

97. 68: How are hypothesis 3 and hypothesis 1 different from one another? Is it because #3 describes growth, whereas #1 is just about biomass as a standing stock? They seem very similar to me.

See revisions under Comment 6

98. 71: Why did you use both ^{13}C and ^{14}C labels? Briefly describe here, and in more detail in the main text. This seems like a key point in your study design, and in what makes it unique, so it would be useful to provide more explanation.

Our use of dual isotopes was purely practical and does not contribute substantially to the novelty of the work in our view. We now explain this in the methods section:

"The ^{14}C label in the added glucose enabled rapid and accurate measurement of glucose-derived C in liquid extracts by scintillation counting" (line 352)

99. 72: Glucose is also common in dissolved organic carbon from root and shoot plant litter.

We have rephrased this sentence to be clear that we are providing non-comprehensive examples:

"...glucose, which is common in soil, including as a component of plant root exudates and the most abundant product of plant-derived organic matter decomposition²⁵." (line 90)

100. 75: What are these amounts based on, in terms of high and low? Reference? Biological scenario?

We have specified this in the revised manuscript:

"Glucose levels were selected to probe the effects of C supply on storage, with additions above and below the magnitude of MBC having potentially contrasting effects on microbial growth⁴⁰." (line 348)

101. 78: Why do such a short incubation? Please explain the justification here for a 96 hour incubation.

See revisions under Comment 76

102. 85: It seems a bit premature to say this study reveals the importance of storage in a natural microbiome, without yet stating how important it is!

We have revised “revealing” to “examining” to emphasize that this was intended as a statement of the purpose of the study, not the outcome.

103. Section 2.1: I think this first section should directly followed from the first hypotheses. The first thing I’d like to see is therefore how much microbial storage compounds account for, in terms of total biomass (i.e. the first hypothesis). This should form the first section and Figure 1, as it is the most critical part of the paper. It is confusing that you begin the first sentence of the results by talking about patterns of soil respiration, and how they align with past observations. How is this relevant to the main point of the paper? Overall, I’m confused how the first three paragraphs tie to the main purpose of this section? Perhaps move these to supplementary, or put later on in the paper? **I’m expecting to hear about storage compounds right off the bat, but this section is about CO2 efflux rates under different nutrient levels.** If the point of this is to show your treatments worked, again, I would put this in supplementary, including your current Figure 1.

We appreciate that the reviewer would like to move as soon as possible to the core results of the paper. However, we feel there is a certain trade-off here. The first section sets out important evidence of the nutritional status of the microbial community during the experiment (see Comment 82). Removing this section would allow the reader to move directly to the core results on storage compounds, but would not provide them with the information to interpret these results fully and appreciate their implications. Therefore we would prefer to clarify the purpose of this section more clearly (Comment 82) but keep it in the main text.

104. Section 2.2: It would be useful here to briefly remind reader how you used isotopes to parse apart different contributions of storage compounds.

We have added this reminder:

“Isotopic composition (¹³C) indicated that assimilation of glucose C ...” (line 167)

105. 129: I found the wording “a pool 25% as large as” to be a confusing turn of phrase. Do you mean it was 25% the size of the extractable biomass pool?

Revised – see Comment 9.

106. 132: “Storage equivalent to a substantial proportion of biomass” Not sure what this means?

This has been revised:

“storage representing a substantial proportion of biomass ...” (line 183)

107. 137: Instead of “widely underestimated” I would just state the numbers ... by about 20 to 45%.

Here we aim to emphasize that these findings challenge the accuracy of a widely used method, and therefore have broad relevance for microbial biomass growth estimates. However, we feel it would be premature to quantitatively extrapolate the estimates from this study to other soils. We would therefore prefer to keep this statement non-quantitative. This is, however, now extended to explicitly state that methodological work is required to address this.

108. 207: Insert some references here?

Done

109. 218: “storage growth’ feels like a confusing term. Maybe ‘allocation to storage compounds’ or something along those lines?

We have revised this to:

“biomass growth through allocation to storage” (line 269)

This should in our opinion be considered a component of growth, as noted in Comment 56. However, we agree that the revision will be clearer for readers to understand our point here.

110. 223: I like this, as it starts to tangibly explain the implications of this study – can you include a bit more detail on how it would necessitate a reassessment of model inputs?

This has been expanded to specifically consider the implications on CUE (see also Comment 12).

“The important model parameter of carbon-use efficiency is typically measured over 24-hour periods³⁵, but over this time-frame we observed storage changes that constituted a substantial component of the microbial C balance. This suggests that more nuanced representations of microbial metabolism and C allocation may be required to accurately account for microbial C use.” (line 276)

111. 234: This is a great, clear paragraph, and would be very useful if it came earlier on – such as in the Introduction!

Thank you for this positive assessment. This phrasing has been incorporated into the introduction:

“This is frequently understood as synonymous with an increase in individuals, in other words, the replicative growth of microbial populations.” (line 37)

112. 242: I appreciate the justification here of conducting such a short incubation, but it would be great if this came earlier. And how might these processes differ over longer time periods than 96 hours, such as sustained conditions of C or N limitation (or other relevant conditions) over the course of a growing season?

Data on storage dynamics over ecologically relevant time-frames is very limited, as the first author highlights in a recent review (Mason-Jones et al., 2021, cited in the manuscript). The current manuscript provides some first insight into the timeframes relevant for C storage. We suspect that it could also be relevant on longer timeframes, and we have added this to the revised manuscript:

“Resource availability in natural and agroecosystems changes over various time-scales, and we hypothesize that microbial storage may also be responsive to, for example, seasonal changes in belowground C inputs, supporting microbial activity through resource-poor winter periods or dry summers.” (line 305)

113. 251: I'm finding Figure 4 a bit confusing, and it feels like its requiring too much time reading a long caption to understand what the figure is supposed to represent. A few thoughts: why does it require the particular X and Y axis? Could it instead directly include the stoichiometric conditions on the figure itself, so the reader doesn't need to refer to the caption? Why are the pie charts necessary? They all include the same ratio of PHB and TAG, so this feels confusing at first, as I'm looking to see if the ratios are different. Its not intuitive to me at first that different sized pie charts indicate different amounts of storage compounds. I think the yellow circles inside the cell do a much better job of this. Are the terms 'storage growth' and 'stoichiometric growth' used elsewhere in the paper? It seems not. I would use these earlier on, clearly define them, and use them throughout, so its very clear to the reader what they are, when they are being used in the final synthesis figure. It could also be useful to contrast the two forms of growth in a conceptual figure like this, by showing equal amounts of growth, but in one instance there are more individuals, and in another there are fewer, but more storage compounds. In this figure, it seems there always the same number of individuals (3).

Thank you for pointing out the difficulties with this figure's complexity. We have simplified it by removing the pie charts as requested, and making it a semi-quantitative figure based on the diagrammatic representation of the cells and intracellular storage. The caption has also be correspondingly shortened. We have retained the axes, as these are key for communicating the two "directions" of biomass growth referred to in the text. However "storage growth" which was causing confusion is now "storage without replication" (see Comment 109). "Stoichiometric growth" is now more clearly explained in the caption. As noted (Comment 95) we would prefer not to place the two modes of storage at centre stage, since we see these as powerful interpretative tools but not the core message of this paper.

114. 267-270: I think you need a longer, more in depth discussion of the implications of this work. It could also be useful to discuss some of the limitations of this study, and key next steps.

See Comments 81 and 86.

REVIEWERS' COMMENTS

Reviewer #1 (Remarks to the Author):

Considering the reviewer's comments, the authors have improved the manuscript. I think the authors have addresses the reviewer's concerns and I don't see any further issues. I highly encourage the publication of the manuscript.

Reviewer #2 (Remarks to the Author):

The authors made substantial revisions to the manuscript in response to the comments from three reviewers. These changes have significantly improved the readability of the manuscript. The findings of this work are highly important to understanding how microorganisms allocate carbon and resources. The changes the authors made now further support the importance and impact of this work. I am pleased with the revisions and have no further comments.

Reviewer #1 (Remarks to the Author):

Considering the reviewer's comments, the authors have improved the manuscript. I think the authors have addresses the reviewer's concerns and I don't see any further issues.

I highly encourage the publication of the manuscript.

Reviewer #2 (Remarks to the Author):

The authors made substantial revisions to the manuscript in response to the comments from three reviewers. These changes have significantly improved the readability of the manuscript. The findings of this work are highly important to understanding how microorganisms allocate carbon and resources. The changes the authors made now further support the importance and impact of this work. I am pleased with the revisions and have no further comments.

Response:

We thank the reviewers for their kind comments and positive assessment of our work.